# Reticulophagy receptor FAM134C restrains BMP receptor signaling

Shuchen Gu[1,2,3,10] ✉, Hanchenxi Zhang[1,10], Jin Cao[1,2,3], Zhou He[1], Jianfeng Wu[4], Xia Liu[5], Mingjie Zheng [1], Ting Liu [1,9], Bin Zhao [1,3], Pinglong Xu [1,3], Qiming Sun [6,7], Jianping Jin[1,3], Xia Lin[8], Yi Yu[1,2,3], Jiahuai Han[4] & Xin-Hua Feng [1,2,3,7] ✉

## Abstract

FAM134/RETREG family members are ER-phagy receptors that maintain cellular homeostasis by regulating endoplasmic reticulum turnover. However, possible non-ER-phagy functions of FAM134 proteins remain elusive. Here, we show that RETREG3/FAM134C functions as a selective autophagy receptor for the type I BMP receptor (BMPRIA/ALK3) and recruits BMPRIA into LC3-containing autophagosomes for subsequent degradation. FAM134C-induced degradation diminishes the availability of BMP receptors and thus the strength of BMP signaling. Inhibition of autophagy through chemical means or knockdown of key autophagy regulators, ATG5 or Beclin-1, prevents BMPR1A degradation. Additionally, disruption of the putative LC3-interacting region (LIR) motif in FAM134C completely abolishes its interaction with LC3, thereby impeding its ability to degrade BMPR1A. Moreover, FAM134C-deficient mice exhibit enhanced BMP responses in the intestines, which affects intestinal crypt regeneration. Our findings suggest that FAM134C acts as a specific receptor that controls BMP signaling through the autophagic degradation of the type I BMP receptor, independent of its canonical role in ER-phagy.

**Keywords** RETREG; TGF-β; Smad; Autophagy; Degradation
**Subject Categories** Autophagy & Cell Death; Digestive System; Organelles

## Introduction

The endoplasmic reticulum (ER) can undergo selective autophagy, known as ER-phagy, in response to autophagy induction and/or ER stress signals. This process plays a critical role in maintaining cellular homeostasis and quality control (Chino and Mizushima, 2020; Gubas and Dikic, 2022; Reggiori and Molinari, 2022). The FAM134/RETREG reticulum protein family, comprising FAM134A, FAM134B, and FAM134C that resides in the ER, plays a crucial role in ER-phagy (Khaminets et al, 2015; Reggio et al, 2021). These proteins recognize and target damaged or excess ER for degradation. Among the family, FAM134B is the first member to be shown to facilitate ER degradation (Khaminets et al, 2015) and the best described in its physiological and pathological roles (Chino and Mizushima, 2020; Gubas and Dikic, 2022; Reggiori and Molinari, 2022). While the three FAM134 members share overlapping functions, FAM134B appears to act under basal conditions; FAM134A and FAM134C also interact with the autophagy modifier LC3 and may drive ER-phagy in response to stress stimuli (Khaminets et al, 2015; Reggio et al, 2021). In addition to ER protein turnover, FAM134 members control degradation of Collagen-I; yet intriguingly, FAM134C may act as a co-receptor together with the other two family members in this process. Nonetheless, the precise role of FAM134C remains poorly understood (Khaminets et al, 2015; Reggio et al, 2021). It is unknown whether and how any FAM134 members can mediate specific degradation of non-ER proteins, especially signaling molecules or pathways. Thus, understanding the functions of the FAM134/RETREG family in regulating cellular signaling can provide valuable insights into its implications for various physiological responses and diseases.

Bone morphogenetic proteins (BMPs), belonging to the TGF-β superfamily, were first identified for their important role in bone and cartilage formation 50 years ago. Recent studies have underscored the broad significance of BMP signaling in regulating various cellular functions (Akiyama et al, 2024; Miyazono et al, 2010; Nohe et al, 2004; Plouhinec et al, 2011; Wozney et al, 1988; Wu et al, 2024; Zhou et al, 2023). Emerging evidence suggests that BMPs are key players in maintaining tissue homeostasis in the adult cardiovascular, respiratory, digestive, and nervous systems (Akiyama et al, 2024; Bier and De Robertis, 2015; Wu et al, 2024; Zhou et al, 2023). Deregulation of BMP signaling can lead to developmental defects and human diseases. The precise outcomes of BMP signaling are finely regulated by controlling the availability and activity of ligands, receptors, and other regulators.

BMPs, like other TGF-β family members, interact with type I and type II serine-threonine kinase receptors. The BMP type II

[1]MOE Key Laboratory of Biosystems Homeostasis & Protection and Zhejiang Key Laboratory of Molecular Cancer Biology, Innovation Center for Cell Signaling Network, Life Sciences Institute, Zhejiang University, Hangzhou, Zhejiang 310058, China. [2]Center for Life Sciences, Shaoxing Institute, Zhejiang University, Shaoxing, Zhejiang 321000, China. [3]Cancer Center, Zhejiang University, Hangzhou, Zhejiang 310058, China. [4]State Key Laboratory of Cellular Stress Biology, Innovation Center for Cell Biology, School of Life Sciences, Xiamen University, Xiamen, Fujian 361005, China. [5]ZJU-Hangzhou Global Scientific and Technological Innovation Center, Zhejiang University, Hangzhou, Zhejiang 311215, China. [6]International Institutes of Medicine, The Fourth Affiliated Hospital, Zhejiang University, Yiwu, Zhejiang 322000, China. [7]The Second Affiliated Hospital, Zhejiang University, Hangzhou, Zhejiang 310009, China. [8]Department of Hepatobiliary and Pancreatic Surgery and Zhejiang Provincial Key Laboratory of Pancreatic Disease, The First Affiliated Hospital, Zhejiang University School of Medicine, Hangzhou, Zhejiang 310003, China. [9]Present address: School of Basic Medicine, Zhejiang University, Hangzhou, Zhejiang 310058, China. [10]These authors contributed equally: Shuchen Gu, Hanchenxi Zhang. ✉E-mail: fenglab@zju.edu.cn; xhfeng@zju.edu.cn

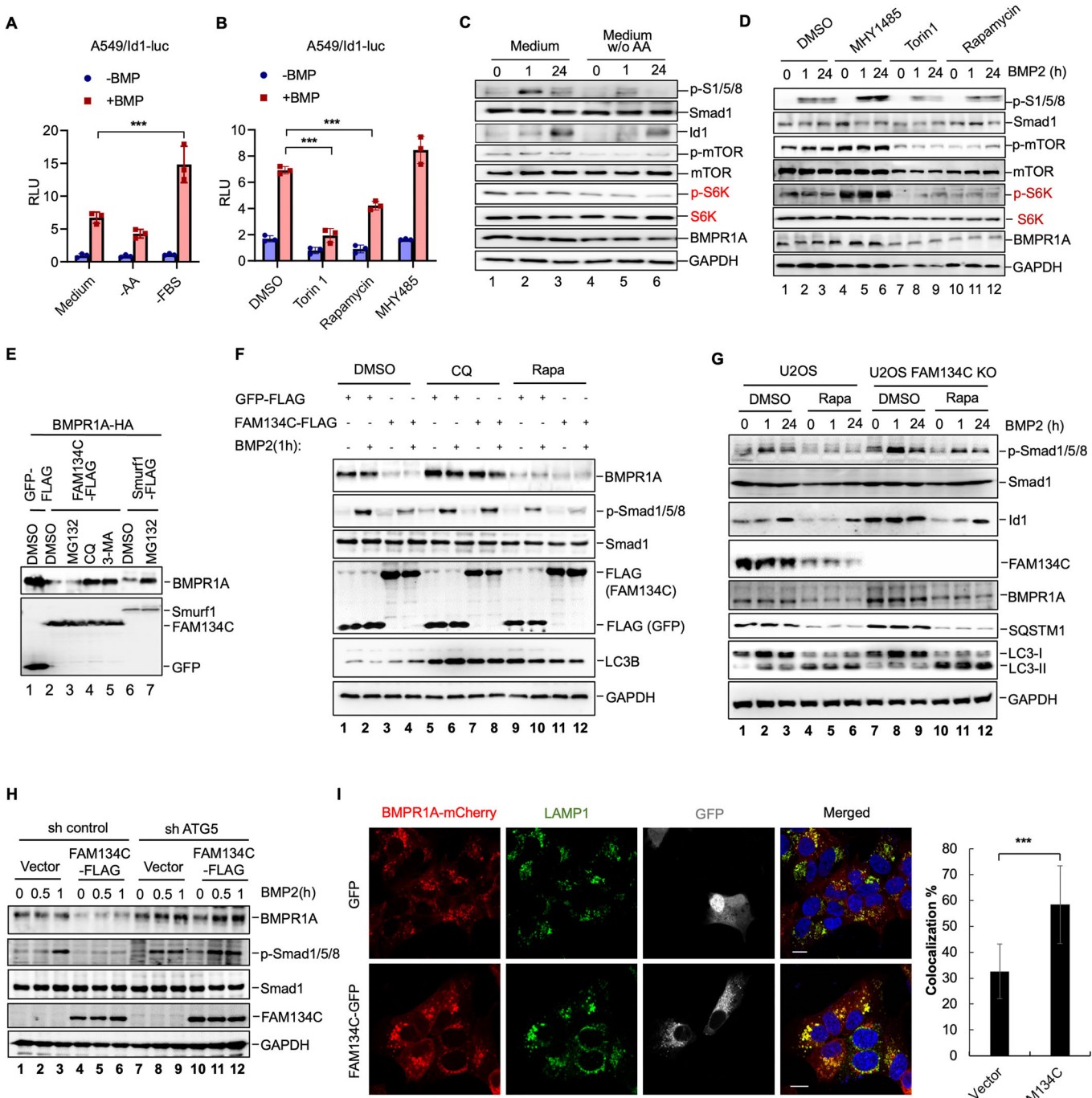

receptor (BMPRII or ActRIIB) is responsible for phosphorylating the type I BMP receptor, primarily BMPR1A or BMPR1B. BMPR1 governs signaling specificity, and once activated, it phosphorylates receptor-regulated Smads (R-Smads), i.e., Smad1, 5, and/or 8. Upon forming a heterocomplex with Smad4, activated R-Smads become accumulated in the nucleus where they regulate the transcription of target genes such as Id1 and Id2 (Akiyama et al, 2024; Sanchez-Duffhues et al, 2020; Sieber et al, 2009; Wu and Hill, 2009). The transcriptional activity of Smads is regulated by numerous coactivators and corepressors within the nucleus (Hill, 2016).

Moreover, BMP signaling is modulated by extracellular antagonists like Noggin, Chordin, Follistatin, Tsg, and Dan family members, which inhibit BMP binding to its receptor (Todd et al, 2020). Intracellularly, inhibitory Smads (including Smad6 and Smad7) exert negative regulation on BMP signaling through diverse mechanisms (Bai and Cao, 2002; de Ceuninck van Capelle et al, 2020; Yan et al, 2009), including the recruitment of E3 ligases to degrade receptors and Smads or phosphatases to dephosphorylate receptors (Liu et al, 2022; Murakami et al, 2003). HECT-type E3 ubiquitin ligases, namely Smurf1 and Smurf2, prevent excessive

**Figure 1.    FAM134C promotes autophagic degradation of BMPR1A.**

(A) Amino acids withdrawal attenuates BMP-induced Id1-luc reporter activity. A549 cells were transfected with Id1-luc reporter plasmids, after 24 h change medium to complete culture medium or nutrient depleted medium (without amino acids or FBS) with or without BMP2 (50 ng/ml) for 8 h. Relative luciferase activity was determined using a microplate-type luminometer with luciferase assay reagent. Statistical analysis by unpaired two-tailed Student's $t$-test; *$p < 0.05$, **$p < 0.01$, ***$p < 0.001$; mean ± SD ($n = 3$ independent experiments). After BMP stimulation, Medium vs -FBS $p = 8.39\text{E-}03$, (B) mTOR inhibition attenuates BMP-induced Id1-luc reporter activity. A549 cells were transfected with Id1-luc reporter plasmids, after 24 h cells stimulated with Torin1, Rapamycin or MHY1486 and together with or without BMP2 for 8 h. Relative luciferase activity were determined using a microplate-type luminometer with luciferase assay reagent. Statistical analysis by unpaired two-tailed Student's $t$-test; *$p < 0.05$, **$p < 0.01$, ***$p < 0.001$; mean ± SD ($n = 3$ independent experiments). After BMP stimulation, DMSO vs Torin1 $p = 4.34\text{E-}04$; DMSO vs Rapamycin $p = 1.35\text{E-}04$, (C) Amino acids withdrawal blocks BMP signaling. A549 cells were cultured with complete culture medium or amino acids withdrawal medium for 4 h, then treated with BMP2 (50 ng/ml) for 1 or 24 h. Expression of p-Smad1/5/8, Smad1, Id1, p-mTOR, mTOR, p-S6K, S6K, BMPR1A, and GAPDH were measured by Western blotting. (D) mTOR inhibition blocks BMP signaling. A549 cells were stimulated with Torin1, Rapamycin or MHY1485 and together with or without BMP2 for 1 or 24 h. Expression of p-Smad1/5/8, Smad1, p-mTOR, mTOR, p-S6K, S6K, BMPR1A, and GAPDH were measured by Western blot. (E) FAM134C degrades BMPR1A by autophagy. HEK293T cells were co-transfected with vectors encoding FAM134C-FLAG or Smurf1-FLAG and BMPR1A-HA, 24 h after transfection, cells were treated with MG132 (20 µM), CQ (10 µM), or 3-MA (200 µM) for 4 h, and expression of FAM134C, Smurf1 and BMPR1A were measured by Western blotting. (F) FAM134C promotes BMPR1A degradation and inhibits Smad1/5/8 phosphorylation. U2OS cells were stably expressing FAM134C-FLAG or GFP-FLAG as a control, pretreated with CQ (10 µM) or Rapamycin (100 nM) for 4 h, and then treated with BMP2 (50 ng/ml) for 1 h. Expression of BMPR1A, p-Smad1/5/8, Smad1, FAM134C, LC3, and GAPDH were measured by Western blotting. (G) FAM134C knockout stabilizes BMPR1A. U2OS or U2OS FAM134C knockout cells were treated with Rapamycin (100 nM) and BMP2 (50 ng/ml) for 1 or 24 h. Expression of BMPR1A, p-Smad1/5/8, Smad1, FAM134C, LC3, and GAPDH were measured by Western blotting. (H) ATG5 knockdown blocks FAM134C-dependent degradation of BMPR1A. ATG5 stable knockdown U2OS cells were transfected with FAM134C-FLAG or the FLAG vector. Twenty-four hours after transfection, cells were treated with BMP2 (50 ng/ml) for 0.5 or 1 h. Expression of BMPR1A, p-Smad1/5/8, Smad1, FAM134C, and GAPDH were measured by Western blotting. (I) FAM134C is colocalized with BMPR1A in lysosomes. U2OS cells were co-transfected with GFP-FAM134C and mCherry-BMPR1A. After treatment with CQ (10 µM) 4 h, cells were analyzed by immunofluorescence for FAM134C and BMPR1A, and lysosomes (labeled by an anti-LAMP1 antibody) using Zeiss LSM880. DAPI (blue fluorescence) was used to stain nuclei (scale bar, 10 µm). Percentage of FAM134C colocalized with BMPR1A in the lysosome was quantified by ImageJ. Statistical analysis by unpaired two-tailed Student's $t$-test; *$p < 0.05$, **$p < 0.01$, ***$p < 0.001$; mean ± SD ($n = 3$ independent experiments; each with five technical replicates). Vector vs FAM134C $p = 1.00\text{E-}02$. Source data are available online for this figure.

activation of BMP signaling via ubiquitination-mediated proteasomal degradation of the type I receptor (Liu et al, 2022). Protein degradation through lysosome-autophagy is also a critical cellular process that helps maintain cellular homeostasis. Yet, it remains elusive whether lysosome-dependent degradation regulates BMP signaling.

In this study, we have uncovered RETREG3/FAM134C as a novel negative regulator of BMP signaling through the autophagy-lysosome pathway. Our findings demonstrate that FAM134C interacts with BMPR1A, sequestering the latter into autophagosomes for subsequent degradation. Disruption of the LC3-interacting region (LIR) motif in FAM134C or inhibition of autophagy prevented BMPR1A degradation. Notably, increased BMPR1A levels profoundly heightened BMP responses in the intestines of FAM134C−/− mice, resulting in decreased organoid formation in vitro and delayed intestinal crypt regeneration in vivo. In conclusion, our findings establish FAM134C-mediated autophagolysosomal degradation of BMP type I receptor BMPR1A as a critical mechanism for fine-tuning BMP signaling.

## Results

### FAM134C promotes autophagic degradation of BMPR1A

In our attempt to investigate the potential regulation of BMP signaling under stress conditions, we inadvertently discovered that withdrawal of amino acids led to a clear reduction in BMP-induced Id1-luc reporter expression, while absence of fetal bovine serum (FBS), as a control, resulted in increased Id1-luc activation (Fig. 1A). Given that mTORC1 can be activated by nutrients such as amino acids, we further evaluated the activity of mTOR in regulating BMP responses. The results revealed that the application of mTOR inhibitors Torin1 or Rapamycin diminished Id1-luc reporter activity, whereas the mTOR activator MHY1485

augmented the BMP response (Fig. 1B). In accordance, amino acid withdrawal, which inhibited mTOR activation-mediated S6K phosphorylation, hindered BMP-induced phosphorylation of Smad1/Smad5 (Fig. 1C). Direct mTOR inhibition using Torin1 and Rapamycin reduced, while MHY1485 increased, Smad1/Smad5 phosphorylation (Fig. 1D). These findings suggest that the nutrient status may influence the function of upstream BMP receptors. Furthermore, our observations indicate that mTOR inhibition indeed reduced the expression of the type I BMP receptor BMPR1A (also known as ALK3) (Fig. 1C,D).

To further investigate how the BMP receptor signaling is regulated, we conducted a protein-protein interaction screen using BMPR1A as a bait in immunoprecipitation-coupled mass spectrometry assays. As a result, we identified FAM134C, a protein involved in ER-phagy (Di Lorenzo et al, 2022), as a novel BMPR1A-interacting protein. Significantly, transient expression of FAM134C substantially reduced the protein level of BMPR1A in cells (Fig. 1E, lanes 2–3), although FAM134C did not change the mRNA levels of BMPR1A (Fig. EV1A–D). Similarly, overexpressed FAM134C promoted the degradation of ACVR1/ALK2, ACVRL1/ALK1, and to a lesser extent, BMPR1B/ALK6. In addition, FAM134C reduced the level of BMPRII, presumably through its complex with the type I receptor (Fig. EV1E). We then asked how FAM134C decreases the protein level of type I receptor BMPR1A. As shown in Fig. 1E, the decreased BMPR1A protein level induced by FAM134C overexpression was restored upon treatment with autophagy-lysosomal inhibitor chloroquine (CQ) or 3-MA, but not with proteasome inhibitor MG132 (Fig. 1E). In contrast, MG132 could clearly reverse the effect of Smurf1-mediated BMPR1A degradation (Fig. 1E). This indicates that FAM134C may facilitate the degradation of BMPR1A through autophagy-lysosomal degradation. Furthermore, in cells stably expressing FAM134C, we also observed an apparent decrease in endogenous BMPR1A levels compared to those in the control group in U2OS cells (Fig. 1F, compare lanes 3–4 to 1–2). Notably, treatment with CQ inhibited

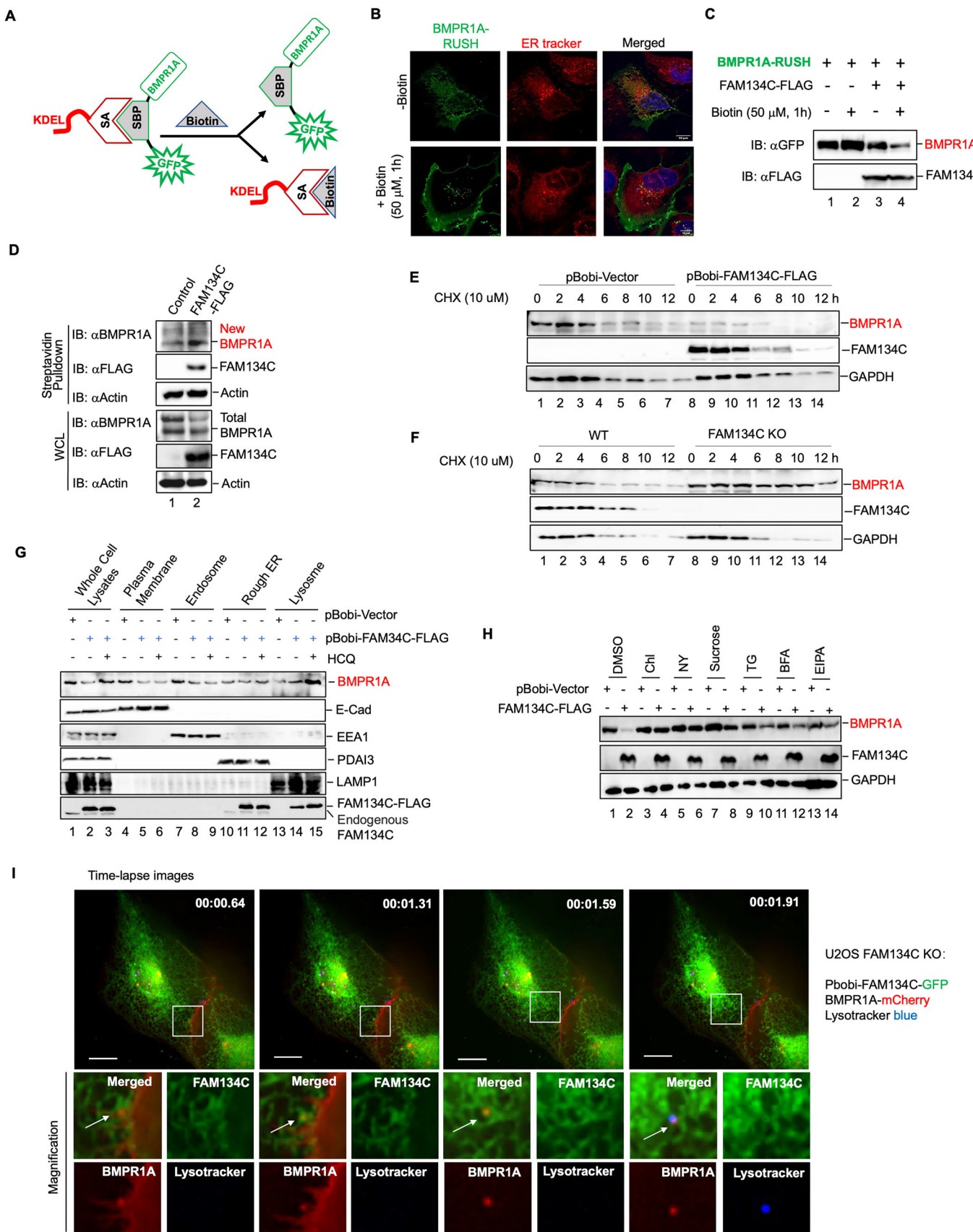

(A) Scheme diagram of the RUSH system. The RUSH system consists of two expression plasmids: KDEL-SA, an ER hook (KDEL) fused to streptavidin (SA); BMPR1A-SBP-GFP, BMPR1A linked to green fluorescent protein (GFP) through a streptavidin-binding peptide (SBP). (B) BMPR1A is localized on the plasma membrane. Immunofluorescence images show the trafficking of BMPR1A from the ER in the absence of biotin to the plasma membrane in the presence of biotin (50 µM, 1 h). Cells were analyzed using Zeiss LSM880. BMPR1A is marked with BMPR1A-RUSH (green fluorescence). ER tracker (red fluorescence) was used to stain the ER. DAPI (blue fluorescence) was used to stain nuclei (scale bar, 10 µm). (C) FAM134C degrades BMPR1A. HEK293T cells were co-transfected with empty vector (negative control) or FAM134C-FLAG and BMPR1A-RUSH. Twenty-four hours after transfection, cells were treated with or without Biotin 50 µM for 1 h, and expression of FAM134C and BMPR1A were detected by Western blotting. (D) FAM134C does not degrade newly synthesized BMPR1A. BONCAT assay in A549 cells stably expressing empty vector (negative control) and FAM134C-FLAG. The newly synthesized protein was precipitated by Streptavidin agarose beads. Endogenous BMPR1A expression was detected by Western blotting. (E) FAM134C overexpression causes BMPR1A degradation. U2OS cells stably expressing FAM134C-FLAG or empty vector control were treated with cycloheximide (CHX, 10 µM) for the indicated times. Expression of BMPR1A, FAM134C, and GAPDH were measured by Western blotting. (F) BMPR1A degradation is blocked in FAM134C knockout cells. FAM134C knockout cells or parental U2OS cells were treated with CHX (10 µM) for the indicated times. Expression of BMPR1A, FAM134C, and GAPDH were measured by Western blotting. (G) FAM134C reduces the levels of BMPR1A on the plasma membrane and endosomes, but not the ER. HCQ blocks BMPR1A degradation in the lysosome. Organelles were collected and analyzed for BMPR1A by Western blotting in FAM134C-expressing A549 cells. Isolation of the plasma membrane, endosome, ER, and lysosome is described in the Materials and Methods section. Expression of BMPR1A, FAM134C, E-cadherin (membrane marker), EEA1 (endosome marker), PDAI3 (ER marker), and LAMP1 (lysosome marker) were measured by Western blotting. (H) FAM134C degrades internalized BMPR1A. U2OS cells stably expressing FAM134C-FLAG or empty vector control were treated with Chlorpromazine (Chl, 50 nM), Nystatin (NY, 50 nM), Thapsigargin (TG, 500 µM), or ethyl-isopropyl amiloride (EIPA, 10 µM) for 15 h, sucrose (0.2 M) and Brefeldin A (BFA, 10 µg/ml) for 2 h. Levels of BMPR1A and FAM134C-FLAG were measured by Western blotting. (I) Live cell image shows that FAM134C targets membrane-bound BMPR1A to lysosomes for degradation. FAM134C-KO U2OS cells stably express FAM134C-GFP and BMPR1A-mCherry. Lysotracker (blue fluorescence) was used to stain the lysosome. Cells were analyzed under Zeiss LSM880 (scale bar, 10 µm). Source data are available online for this figure.

the degradation of BMPR1A caused by FAM134C, whereas rapamycin, an autophagy enhancer, profoundly reduced the level of BMPR1A (Fig. 1F). In accordance, the BMP-induced phosphorylation of downstream signal transducers Smad1/5/8 was apparently reduced in FAM134C-overexpressing cells, but this effect was also rescued by CQ treatment (Fig. 1F).

To further confirm the deleterious effect of FAM134C on the BMPR1A stability, we constructed U2OS cells by knocking out the FAM134C gene using CRISPR/Cas9 technology. In the FAM134C−/− U2OS cells, the endogenous level of BMPR1A was apparently elevated when compared to that in wild-type U2OS cells (Fig. 1G, compare lanes 7–9 to 1–3). Accordingly, both BMP-induced phosphorylation of Smad1/5/8 and expression of Id1 were increased in FAM134C−/− cells (Fig. 1G). Oppositely, transient expression of FAM134C enabled a significant reduction in the levels of endogenous BMPR1A and p-Smad1/5/8 (Fig. 1H, compare lane 6 to 3).

To provide further evidence that FAM134C degrades BMPR1A through autophagy, we knocked down ATG5 or Beclin1, key components in the autophagy pathway (Fig. EV1F,G), to block autophagy. Notably, FAM134C overexpression no longer resulted in BMPR1A degradation in cells with depletion of ATG5 (Fig. 1H, compare lanes 10–12 to 4–6) or Beclin1 (Fig. EV1H, compare lanes 10–12 to 4–6). Additionally, the phosphorylation of Smad1/5/8 in FAM134C-overexpressing cells was restored upon ATG5 or Beclin1 knockdown (Figs. 1H and EV1H). Furthermore, immunofluorescence analysis revealed a significant increase in lysosomal localization of BMPR1A upon overexpression of FAM134C, while overexpression of GFP did not elicit such an effect (Fig. 1I). Collectively, these findings strongly suggest that FAM134C mediates the degradation of BMPR1A through the autophagy-lysosomal pathway.

## FAM134C targets membrane-bound BMPR1A for degradation

We next aimed to determine the subcellular compartment in which FAM134C degrades BMPR1A. Given that FAM134C plays a crucial

role in ER-phagy (Khaminets et al, 2015; Reggio et al, 2021), it is plausible that it targets ER-synthesized BMPR1A for degradation. To investigate this, we utilized several techniques to differentiate whether FAM134C targets newly synthesized BMPR1A (still in the ER) or mature BMPR1A (localized to the plasma membrane or endosomes). First, we monitored the trafficking of BMPR1A within the secretory pathway using the Retention Using Selective Hooks (RUSH) system (Boncompain et al, 2012; Pacheco-Fernandez et al, 2021). In this assay, BMPR1A was tagged with GFP and fused to a streptavidin-binding peptide (SBP), which was then bound to a streptavidin molecule linked to KDEL (an ER retention signal) (Fig. 2A). In the absence of biotin, BMPR1A remained in the ER; however, in the presence of biotin, the BMPR1A-SBP-GFP complex was released from the ER and transported to its final destination, such as the plasma membrane (Fig. 2B). Notably, we observed that overexpression of FAM134C reduced the levels of BMPR1A-RUSH in the presence of biotin (Fig. 2C). In contrast, FAM134C overexpression had little effect on BMPR1A-RUSH levels in the absence of biotin (Fig. 2C). This suggests that FAM134C promotes the degradation of biotin-released BMPR1A-RUSH at the plasma membrane, rather than the BMPR1A that is retained forcibly in the ER. Second, we employed bio-orthogonal non-canonical amino acid tagging (BONCAT) to label newly synthesized BMPR1A (Fig. EV2A). We found that FAM134C overexpression led to a decrease in total levels of both endogenous and exogenous BMPR1A. However, it did not affect the levels of newly synthesized BMPR1A (Figs. 2D and EV2B,C). Third, following treatment with cycloheximide to inhibit new protein synthesis, we found that BMPR1A degradation was accelerated in cells overexpressing FAM134C, while it was decelerated in FAM134C knockout cells. These findings suggest that FAM134C specifically influences the degradation of the "old" pool of BMPR1A rather than the newly synthesized pool (Figs. 2E,F and EV2D). Furthermore, subcellular fractionation studies revealed that FAM134C degraded BMPR1A localized to the plasma membrane and endosomes, but not within the ER (Figs. 2G and EV2E,F). Treatment with hydroxychloroquine (HCQ) resulted in a significant accumulation of BMPR1A in lysosomes, confirming that FAM134C mediates BMPR1A degradation via lysosomal autophagy (Figs. 2G and EV2E). Additionally, various

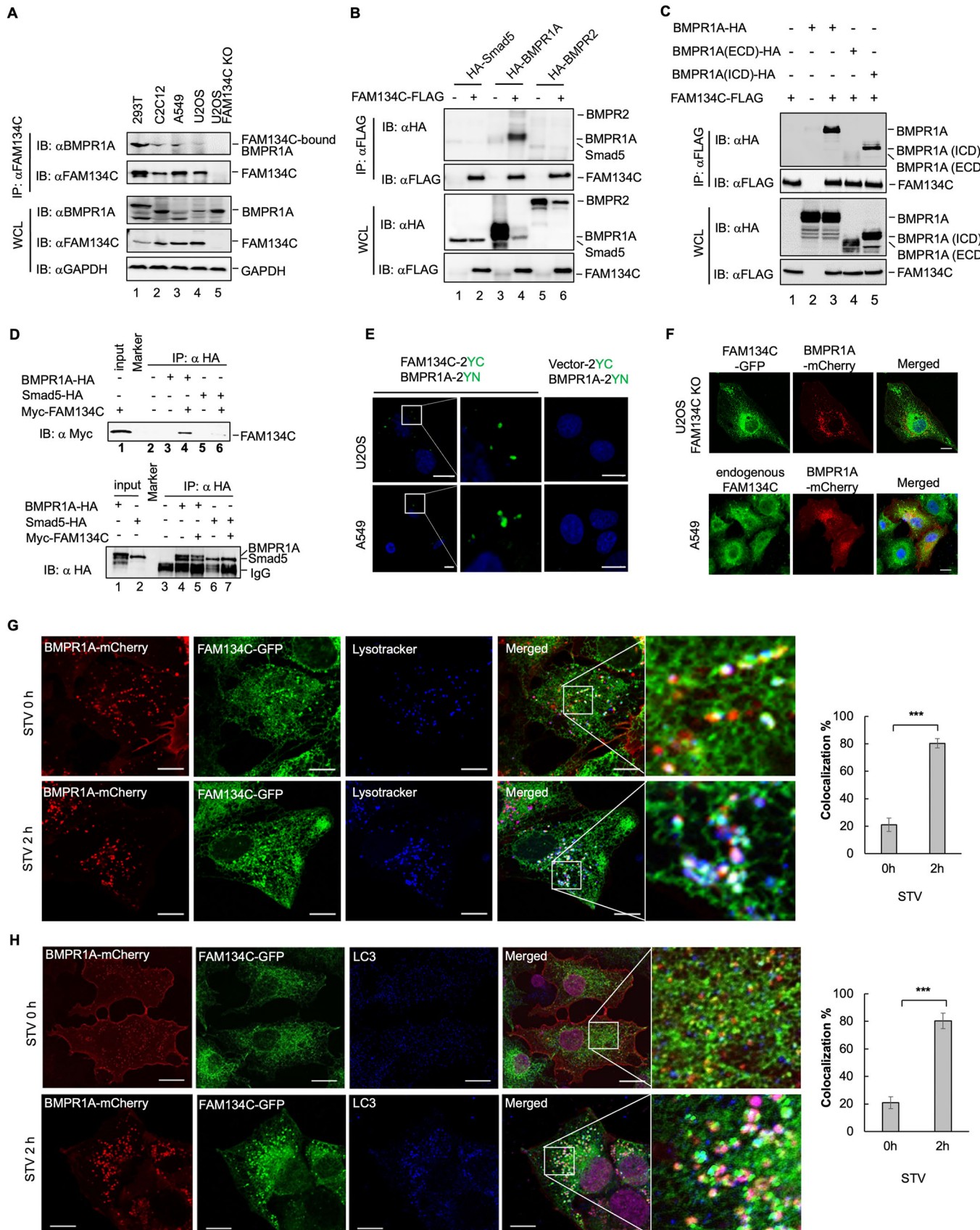

◀ **Figure 3.   FAM134C interacts with BMPR1A under physiological conditions.**

(**A**) FAM134C interacts with BMPR1A at endogenous levels. Endogenous FAM134C was immunoprecipitated by FAM134C antibody in HEK293T, C2C12, A549, U2OS, and FAM134C Knockout U2OS cells, and BMPR1A was detected by Western blotting. (**B**) FAM134C specifically interacts with BMPR1A, but not BMPR2 or Smad5. HEK293T cells were co-transfected with vectors encoding FAM134C-FLAG and Smad5-HA, BMPR1A-HA, or BMPR2-HA, then FAM134C was immunoprecipitated by anti-FLAG antibody, and the protein Smad5, BMPR1A, or BMPR2 was detected by HA antibody. (**C**) FAM134C binds to the intercellular domain of BMPR1A. HEK293T cells were transfected with vectors encoding FAM134C-FLAG and BMPR1A-HA, BMPR1A(ICD)-HA, or BMPR1A(ECD)-HA, then FAM134C was immunoprecipitated by anti-FLAG antibody and BMPR1A or BMPR1A mutation was detected by HA antibody. (**D**) FAM134C interacts with BMPR1A in vitro. MYC-FAM134C, Smad5-HA, and BMPR1A-HA were purified using the TNT in vitro transcription/translation System. Purified Smad5-HA or BMPR1A-HA proteins were incubated with MYC-FAM134C, and then immunoprecipitation was performed using anti-HA antibodies. The IP products were detected by Western blotting with the indicated antibodies. (**E**) FAM134C interacts with BMPR1A in cells. In BiFC assays, A549 or U2OS cells were co-transfected with YC-tagged FAM134C and YN-tagged BMPR1A or YN-tagged vector as a negative control. Cells were analyzed using Zeiss LSM880. The green YFP fluorescence signal indicates the interaction part. DAPI (blue fluorescence) was used to stain nuclei (scale bar, 10 μm). (**F**) Endogenous FAM134C is colocalized with BMPR1A. GFP-tagged FAM134C and mCherry-tagged BMPR1A were expressed in FAM134C-KO U2OS cells (top row), or mCherry-tagged BMPR1A was expressed in A549 cells. These cells were visualized by immunofluorescence under Zeiss LSM880. DAPI (blue fluorescence) was used to stain nuclei (scale bar, 10 μm). (**G, H**) FAM134C is colocalized with BMPR1A in autophagolysosomes under starvation. FAM134C-KO U2OS cells were stably expressed FAM134C-GFP and BMPR1A-mCherry. Cells were starved for 1 h and then analyzed using Zeiss LSM880. Panel (**G**), Lysotracker (blue fluorescence) was used to stain lysosomes (scale bar, 10 μm), and the percentage of FAM134C and BMPR1A colocalized with lysosomes was quantified by ImageJ. Panel (**H**), endogenous LC3 (blue fluorescence) were stained by immunofluorescence (scale bar, 10 μm), and the percentage of FAM134C and BMPR1A colocalized with LC3 were quantified by ImageJ. Statistical analysis by unpaired two-tailed Student's t-test; *$p < 0.05$, **$p < 0.01$, ***$p < 0.001$; mean ± SD. ($n = 3$ independent experiments; each with three technical replicates). 0 h vs 2 h $p = 4.36E-08$(G); 0 h vs 2 h $p = 2.88E-07$(H). Source data are available online for this figure.

inhibitors, Chlorpromazine and Nystatin, as well as sucrose (hypertonic media), effectively blocked FAM134C-mediated degradation of BMPR1A, while Brefeldin A and EIPA did not produce this effect (Figs. 2H and EV2F,G). These results indicate that FAM134C does not impede the transport of newly synthesized BMPR1A but instead negatively influences the levels of internalized BMPR1A. Lastly, through live cell imaging, we discovered that FAM134C targeted membrane-bound BMPR1A (likely from the plasma membrane) to lysosomes for degradation (Figs. 2I; Movie EV1). We conclude that BMPR1A degradation by FAM134C is not mediated through ER-phagy.

## FAM134C physically interacts with BMPR1A

Since FAM134C is a BMPR1A-interacting protein, we sought to validate this interaction using co-immunoprecipitation (co-IP) and in vitro binding experiments. We first observed the endogenous interaction between FAM134C and BMPR1A in various cell lines using a co-IP assay. Apparently, the FAM134C-bound BMPR1A band was specific as it was absent in the U2OS cells with FAM134C knockout (Fig. 3A). In addition, the specific FAM134C-BMPR1A interaction was further demonstrated in co-IP experiments using HEK293T cells transfected with FLAG-tagged FAM134C, HA-tagged BMPR1A, BMPR2, or Smad5. FAM134C effectively interacted with BMPR1A, but not with BMPR2 or Smad5 (Fig. 3B). In co-IP assays, FAM134C interacted with the intracellular domain (ICD) of BMPR1A, but not the extracellular domain (ECD) (Fig. 3C). Furthermore, although ATG5 or BECN1 were required for the FAM134C-induced degradation of BMPR1A (Figs. 1H and EV1G), their depletion did not alter the interaction between FAM134C and BMPR1A (Fig. EV3A).

The FAM134 family has three members: FAM134A, FAM134B, and FAM134C. In comparison to FAM134C, FAM134A did not interact with BMPR1A, while FAM134B displayed a weak interaction with BMPR1A (Fig. EV3B). This supports the notion that the FAM134C-BMPR1A interaction is strong and also specific. Additionally, in vitro-translated FAM134C could bind to in vitro-translated BMPR1A-HA, but not Smad5-HA (Fig. 3D), suggesting a direct interaction between FAM134C and BMPR1A.

We examined the subcellular localization where BMPR1A and FAM134C interacted in living cells using bimolecular fluorescence complementation (BiFC) and immunofluorescence assays. For BiFC, YN-BMPR1A and YC-FAM134C, which the N-terminal (aa 1–154) and C-terminal (aa 155–238) fragments of YFP were fused with BMPR1A and FAM134C, respectively, were transfected into both U2OS and A549 cells. Notably, co-expression of YN-BMPR1A and YC-FAM134C produced distinct fluorescence dots exclusively in the cytoplasm (Fig. 3E). In contrast, the YN-BMPR1A and YC-vector pair did not yield fluorescence in cells. Additionally, immunofluorescence experiments further supported the colocalization of FAM134C with punctate distribution along with BMPR1A in both U2OS and A549 cells (Fig. 3F). Moreover, the interaction between YN-BMPR1A and YC-FAM134C was colocalized with RTN4-mCherry (ER marker), Rab7a-mCherry (late endosome marker) and LAMP1-mCherry (Lysosome marker), suggesting that membrane-bound BMPR1A initially contacted ER-residing FAM134C and the complex ultimately moved towards the lysosomes (Fig. EV3C).

We further determined that BMPR1A and FAM134C were indeed colocalized in lysosomes. By using LysoTracker to mark lysosomes, we observed that co-expression of BMPR1A-mCherry and FAM134C-GFP showed their increased colocalization in lysosomes in live cells following starvation (Fig. 3G). This colocalization was further discovered to increase in autophagosomes marked by LC3 under starvation conditions (Fig. 3H). Importantly, live imaging showed that FAM134C first interacted with BMPR1A and then pulled it into lysosomes, leading to the disappearance of BMPR1A in lysosomes (Fig. EV4; Movie EV2).

## FAM134C recruits BMPR1A into autophagosomes dependently on LC3

As starvation enhanced the localization of FAM134C-BMPR1A with LC3 (Fig. 3H), we were interested in determining their interactions. The FAM134 family proteins have an LC3-interacting region (LIR) that can interact with LC3B and GABARAPL2 (Khaminets et al, 2015). We then assessed the potential interaction between FAM134C and LC3 using BiFC assays. Through their interaction, YN-FAM134C and LC3-YC reconstituted a fluorescent complex in cells (Fig. 4A). We postulated that LC3 proteins help

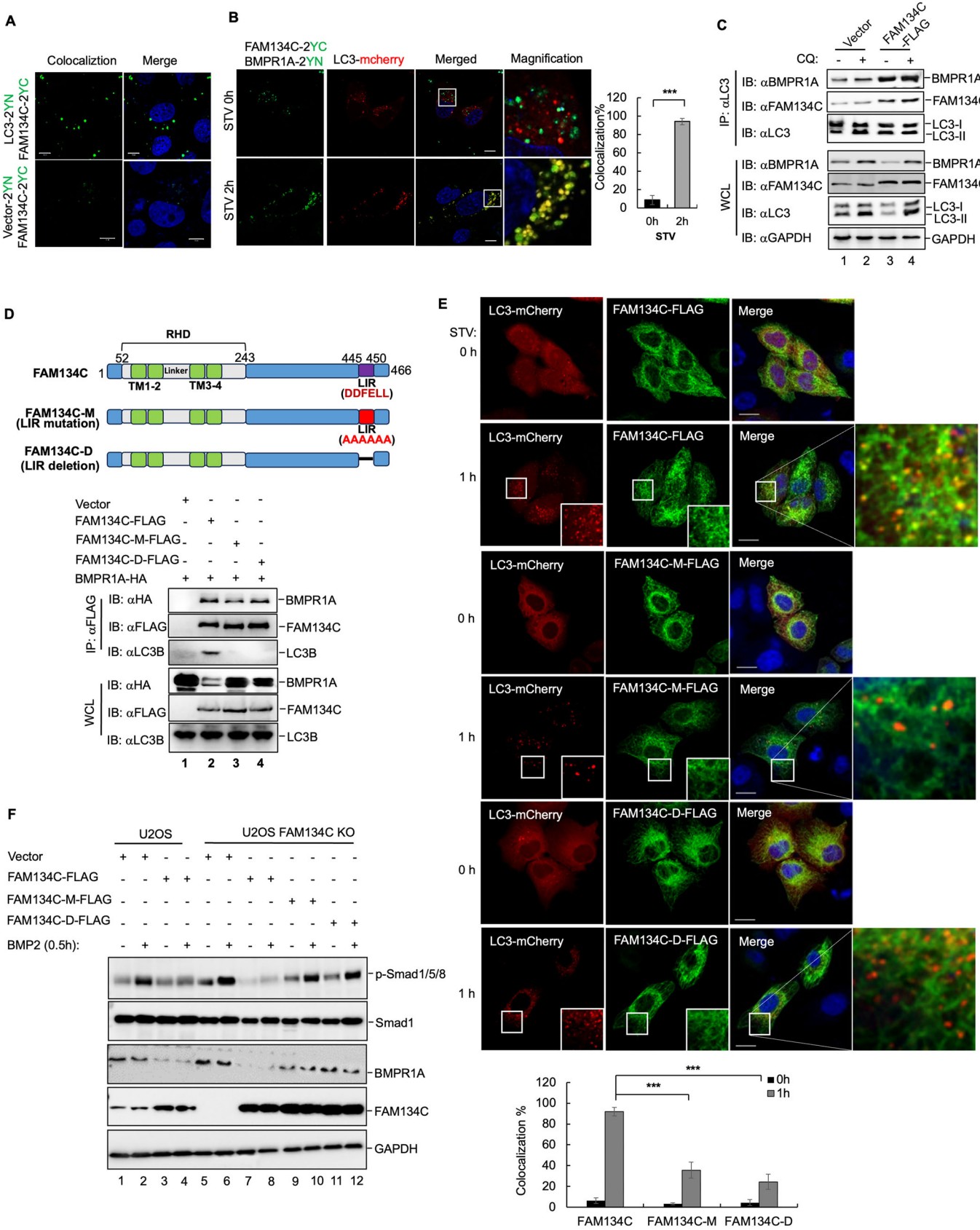

Figure 4. FAM134C targets BMPR1A into autophagosomes through LC3.

(A) FAM134C binds to LC3. In BiFC assays, A549 cells were co-transfected with YC-FAM134C and YN-LC3 or YN-tagged vector as a negative control. Cells were analyzed using Zeiss LSM880. Green YFP fluorescence signal indicates the location of subcellular interaction. DAPI (blue fluorescence) was used to stain nuclei (scale bar, 10 μm). (B) FAM134C interacts with BMPR1A in the LC3-containing autophagosome. U2OS cells were co-transfected with LC3-mCherry with YC-FAM134C and YN-BMPR1A, starved for 2 h, and then analyzed using Zeiss LSM880. Green YFP fluorescence signal indicates the FAM134C-BMPR1A interaction. DAPI (blue fluorescence) was used to stain nuclei (scale bar, 10 μm), and the percentage of FAM134C and BMPR1A colocalized with LC3 were quantified by ImageJ. Statistical analysis by unpaired two-tailed Student's $t$-test; *$p < 0.05$, **$p < 0.01$, ***$p < 0.001$; mean ± SD ($n = 3$ independent experiments; each with three technical replicates). 0 h vs 2 h $p = 1.23E-07$. (C) LC3 is associated with BMPR1A in the presence of FAM134C. U2OS cells were stably expressed with FAM134C-FLAG, treated with CQ (10 μM), and immunoprecipitated by anti-FLAG antibody. LC3-bound BMPR1A or FAM134C were analyzed by Western blotting using the indicated antibodies. (D) FAM134C LIR mutation fails to interact with LC3 and degrade BMPR1A. Top, schematic diagram of domain structures and mutation design of FAM134C. Bottom, HEK293T cells were co-transfected with vectors encoding BMPR1A-HA and FAM134C-FLAG, FAM134C-M-FLAG, or FAM134C-D-FLAG, then FAM134C was immunoprecipitated by anti-FLAG antibody and BMPR1A or LC3 was detected by Western blot. (E) FAM134C, but not its LIR mutants, are colocalized with LC3 after starvation. U2OS cells were co-transfected with GFP-LC3 and FAM134-FLAG, FAM134C-M-FLAG, or FAM134C-D-FLAG. Upon starvation buffer for 1 h, cells were analyzed by fluorescence microscopy using Zeiss LSM880. DAP (blue fluorescence) was used to stain nuclei (scale bar, 10 μm). Bottom, the percentage of FAM134C, FAM134C-M-FLAG, or FAM134C-D-FLAG colocalized with LC3 was quantified by ImageJ. Statistical analysis by unpaired two-tailed Student's $t$-test; *$p < 0.05$, **$p < 0.01$, ***$p < 0.001$; mean ± SD ($n = 3$ independent experiments; each with five technical replicates). FAM134C vs FAM134C-M $p = 1.19E-06$, FAM134C vs FAM134C-D $p = 2.07E-07$. (F) FAM134C LIR mutants fail to degrade BMPR1A. FAM134C Knockout U2OS cells were transfected with FAM134C-FLAG, FAM134C-M-FLAG, or FAM134C-D-FLAG. Twenty-four hours after transfection, cells were treated with BMP2 (50 ng/ml) for 0.5 h. Levels of BMPR1A, p-Smad1/5/8, Smad1, FAM134C, and GAPDH were measured by Western blotting. Source data are available online for this figure.

FAM134C target BMPR1A to lysosomes. To test this, we examined the tripartite colocalization of interacting YN-FAM134C/BMPR1A-YC with mCherry-LC3. YN-FAM134C and BMPR1A-YC formed a fluorescent complex as described (Fig. 3E,B), and importantly, the FAM134C-BMPR1A complex colocalized with LC3-labeled autophagosomes upon starvation (Fig. 4B). Moreover, the presence of FAM134C enabled more endogenous complex between BMPR1A and LC3 (Fig. 4C). These findings indicate that FAM134C degrades BMPR1A by sequestering it into autophagosomes.

To substantiate the requirement of LC3 on FAM134C for BMPR1A degradation, we substituted the amino acid residues of the LIR with alanine (DDFELL to AAAAAA; FAM134C-mLIR-FLAG) or deleted the LIR (FAM134C-ΔLIR-FLAG) (Fig. 4D, top). Either mutant of FAM134C failed to interact with LC3 (Fig. 4D, bottom). As expected, these two mutations abolished their localizations to autophagosomes during autophagy (Fig. 4E). Although they retained the ability to interact with BMPR1A (Fig. 4D), both mutants lost their activity to degrade the receptor (Fig. 4D). In FAM134C−/− U2OS cells, overexpression of these two mutants disabled the function of FAM134C to degrade BMPR1A (Fig. 4F, compare lanes 10 & 12 with 8). These data strongly suggest that FAM134C degrades BMPR1A by sequestering it into autophagosomes, and the LIR plays a crucial role in this process.

## FAM134C attenuates BMP-induced transcriptional responses and intestinal tissue homeostasis

Having demonstrated that FAM134C promotes BMPR1A degradation, we next investigate how FAM134C affects BMP transcriptional responses. We examined the impact of FAM134C on the expression of the canonical BMP target genes. As shown in Fig. 5A, stable expression of FAM134C led to a reduction in the BMP-mediated induction of Id1 and Id2 mRNA levels, and attenuated the BMP4-induced expression of Id1 protein (Fig. EV5A). Conversely, in FAM134C-/- U2OS cells, the BMP2-mediated induction of Id1 and Id2 mRNA levels as well as protein levels was profoundly increased (Figs. 5B and EV5A). Additionally, the level of BMP-induced phosphorylation of Smad1/5/8 was consistent with that of BMPR1A in the presence or absence of FAM134C (Fig. EV5A).

To better understand FAM134C's role in BMP signaling in vivo, we generated Fam134C knockout mice. Since BMP signaling plays an important role in the colonic/intestinal epithelia (McCarthy et al, 2020; Wang and Chen, 2018; Zhang and Que, 2020), we first examined whether Fam134C depletion alters epithelial homeostasis. In Fam134C−/− mice, the protein levels of BMPR1A and p-Smad1/5/8, which are normally observed only at the top of the villi, could now be detected in the crypt bases in both small intestines (Fig. 5C) and colons (Fig. EV5B). Furthermore, BMPR1A protein levels were accordingly elevated in the Fam134C−/− mice (Fig. 5D). These data suggest that loss of Fam134C disrupted the physiological level of BMPR1A and Smad1/5/8 phosphorylation in the epithelium, similarly to what we observed in cultured cells. The data also revealed correspondingly reduced expression of the intestinal stem cell marker, Sox9, in the crypt bases of small intestines (Fig. 5E) and colon (Fig. EV5B), accompanied by a slight increase in the levels of intestinal epithelial markers such as goblet (Muc2) and enteroendocrine lineages (ChgA) (Fig. 5C) in small intestines. In the colon, no significant changes in the differentiation of goblet and enteroendocrine lineages were observed in Fam134C knockout mice (Fig. EV5C), yet deletion of Fam134C caused an increased expression of Id1 and Id2 (Fig. EV5D).

To investigate the relationship between BMPR1A levels and autophagic degradation in vivo, we examined the impact of nutrient deprivation on BMPR1A levels in the intestines of Fam134C knockout mice. Through immunofluorescence staining and Western blotting, we found that fasting led to a significant reduction in BMPR1A levels, along with a decrease in the autophagy protein p62/SQSTM1, in the intestinal tissues of wild-type mice (Figs. 5F,G and EV5E). In sharp contrast, a notable level of BMPR1A was detected in the fasting Fam134C−/− intestines, while p62 levels remained comparable to those in wild-type intestines (Figs. 5F and EV5E). These results indicate that Fam134C deficiency specifically protects BMPR1A, without affecting other autophagy proteins.

We next asked whether over-activation of the BMP response in the intestine would affect the rate of intestinal regeneration after irradiation challenge. Intestinal epithelial cells are sensitive to ionizing radiation but can recover efficiently from low-dose radiation (Bach et al, 2000; Blanpain et al, 2011). Stem cells are

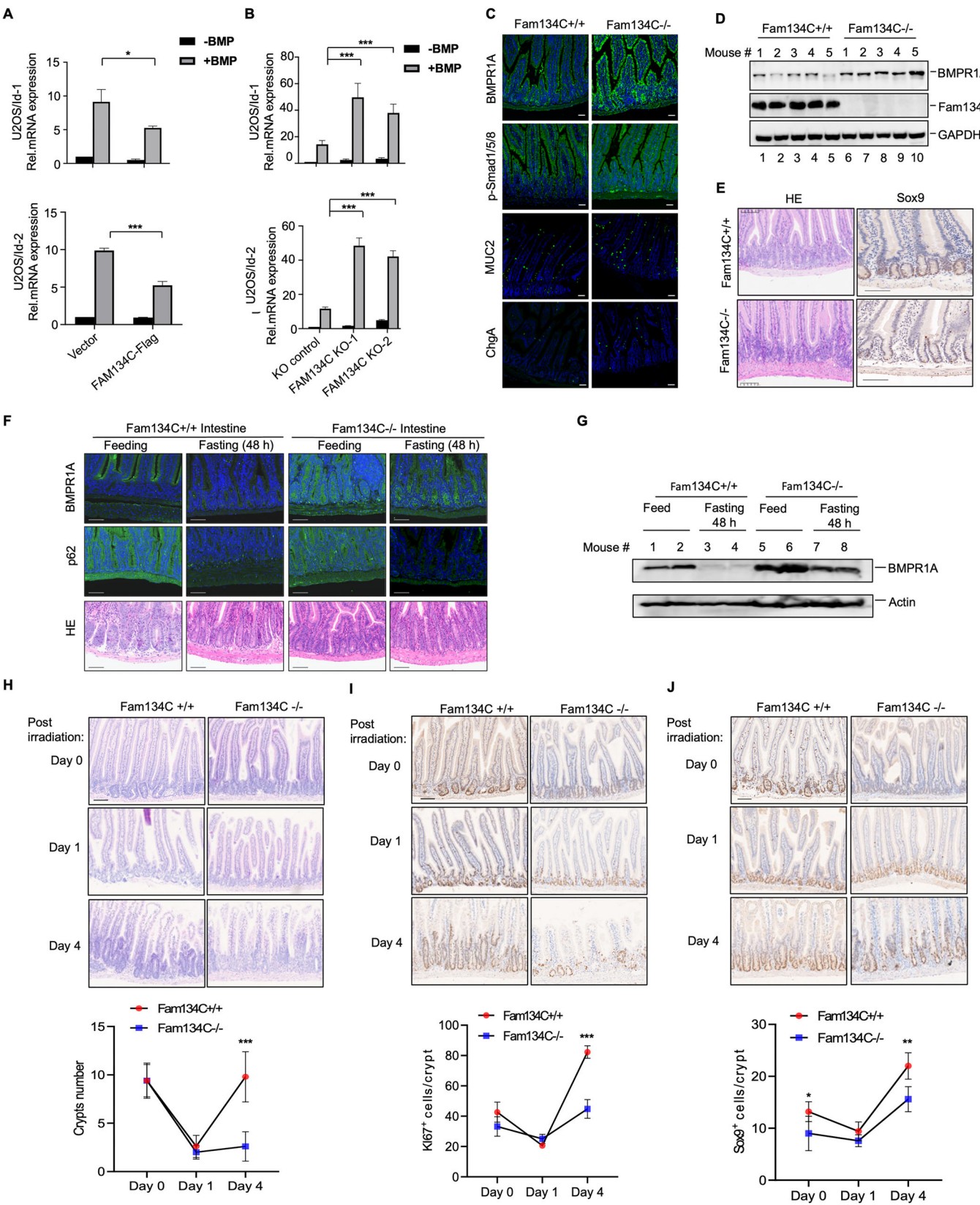

**Figure 5. FAM134C blocks BMP signaling.**

(A) FAM134C inhibits BMP-induced Id1 and Id2 mRNA transcription. U2OS cells stably expressing FAM134C-FLAG or empty vector control) were treated by BMP (50 ng/ml) for 12 h, Id1 and Id2 mRNA were measured by qPCR. Statistical analysis by unpaired two-tailed Student's $t$-test; $*p < 0.05$, $**p < 0.01$, $***p < 0.001$; mean ± SD ($n = 3$ independent experiments). In Id1 mRNA expression, treated with BMP, Vector vs FAM134C-Flag $p = 2.20E-02$. In Id2 mRNA expression, treated with BMP, Vector vs FAM134C-Flag $p = 1.76E-04$. (B) Knockout of FAM134C increases BMP-induced Id1 and Id2 mRNA transcription. WT or FAM134C-KO U2OS cells were treated by BMP (50 ng/ml) for 12 h, Id1 and Id2 mRNA were measured by qPCR. Statistical analysis by unpaired two-tailed Student's $t$-test; $*p < 0.05$, $**p < 0.01$, $***p < 0.001$; mean ± SD ($n = 3$ independent experiments). In Id1 mRNA expression, treatment with BMP, KO control vs FAM134C-KO-1 $p = 5.01E-03$, KO control vs FAM134C-KO-2 $p = 4.71E-03$. In Id2 mRNA expression, treatment with BMP, KO control vs FAM134C-KO-1 $p = 1.62E-04$, KO control vs FAM134C-KO-2 $p = 1.12E-04$. (C) BMPR1A is increased in the small intestine of FAM134C$-/-$ mice. Immunofluorescence staining of BMPR1A, p-Smad1/5/8, MUC2, and ChgA were done in the proximal jejunum of WT and KO mice at week 8. DAPI (blue fluorescence) was used to stain nuclei (scale bar, 50 μm). Images are representative of $n = 6$ mice per genotype. (D) FAM134C deficiency increases BMPR1A expression in mouse intestines. WB analysis of BMPR1A, FAM134C, and GAPDH in the colon of WT and FAM134C knockout mice ($n = 5$ mice per genotype). (E) Hematoxylin-eosin staining and immunohistochemical staining of Sox9 in the small intestines of WT and FAM134C knockout mice. Proximal jejunum sections were made at week 8 (scale bar, 100 μm). Images are representative of six mice per genotype. (F, G) FAM134C deficiency increases the level of BMPR1A in the mouse intestines during fasting. (F) Immunofluorescence staining of BMPR1A and p62, and hematoxylin-eosin staining were done in the small intestines of WT and FAM134C knockout mice with or without fasting. Proximal jejunum sections were harvested at week 8. DAPI (blue fluorescence) was used to stain nuclei (scale bar, 100 μm). (G) Analysis of BMPR1A expression in the small intestines of WT and FAM134C knockout mice with or without fasting ($n = 5$ mice per genotype) was done by Western Blotting. (H–J) Loss of FAM134C hampers crypt regeneration after exposure to ionizing radiation. Proximal jejunum sections were collected at different time points in WT and FAM134C knockout mice after 12 Gy abdominal X-ray radiation ($n = 5$ mice per genotype). (H) Haematoxylin-eosin staining, representative images with quantification of crypt number. (I) Representative immunohistochemical staining of Ki67 and its quantification of Ki67+ cells. (J) Representative immunohistochemical staining of Sox9 with quantification of Sox9+ cells. Statistical analysis by unpaired two-tailed Student's $t$-test; $*p < 0.05$, $**p < 0.01$, $***p < 0.001$; mean± SD. of $n = 5$ mice per genotype (scale bar, 50 μm). Crypts number in day 4 WT vs KO $p = 6.72E-04$; KI67$^+$ cell/crypt in day 4 WT vs KO $p = 3.31E-06$; Sox9$^+$ cell/crypt in day0 WT vs KO $p = 4.00E-02$, Sox9$^+$ cell/crypt in day4 WT vs KO $p = 3.53E-03$. Source data are available online for this figure.

highly susceptible to DNA damage from irradiation, and BMPs play a pivotal role in maintaining the balance between homeostatic self-renewal and proliferation of intestinal epithelial cells. BMP signaling inhibits intestinal stem cell self-renewal by promoting differentiation through Smad-mediated transcriptional activation of differentiation genes and suppression of stemness factors like Lgr5. This pathway acts in opposition to Wnt signaling, with high activity in villus regions driving enterocyte maturation while BMP antagonists (e.g., Noggin) in crypt niches maintain stem cell pools. Disrupted BMP signaling leads to pathological stem cell expansion, as seen in polyposis syndromes, demonstrating its critical role in intestinal homeostasis (McCarthy et al, 2020; Wang and Chen, 2018; Zhang and Que, 2020). Consistent with previous findings (Hagemann et al, 1971; Hua et al, 2012; Withers and Elkind, 1970), we observed that 10 Gy X-ray irradiation led to the loss of crypts in both wild-type (WT) and Fam134C$-/-$ mice on day 1, followed by an increase in crypts in WT mice on day 4 (Fig. 5H). However, Fam134C$-/-$ mice exhibited a limited number of stem cells after irradiation, resulting in a significantly delayed regenerative response, and Fam134C$-/-$ mice regenerated fewer crypts on day 4 after IR irradiation (Fig. 5H). Moreover, the number of Ki67- and Sox9-positive cells in each regenerating crypt was significantly reduced in Fam134C$-/-$ mice compared to WT mice at 4 days after irradiation (Fig. 5I,J). These findings suggest that Fam134C$-/-$ mice have impaired stem cell regeneration following radiation injury.

## FAM134C deficiency inhibits intestinal stem cell maintenance

To further investigate the impact of FAM134C on intestinal homeostasis and regeneration through BMP signaling, we utilized an in vitro culture system of intestinal organoids that requires the ENR medium containing epidermal growth factor (EGF), Noggin, and R-spondin1 (Sato et al, 2009). Intestinal organoids are derived from isolated crypts containing stem cells, and under normal conditions, they develop new crypt domains by budding in the ENR medium. As expected, we

observed that control organoids expanded and generated numerous crypt buds within 4 days of culture (Fig. 6A). In contrast, organoids from Fam134C$-/-$ mice exhibited impaired growth, regression, and a significant number of cell deaths compared to the controls. Notably, the number of buds per organoid was substantially reduced in Fam134C$-/-$ organoids, with ~80% of them showing no buds at all, in contrast to fewer (~15%) observed in controls (Fig. 6A).

Furthermore, the levels of intestinal stem cell marker Lgr5 in Fam134C$-/-$ organoids were lower than those in the control organoids (Fig. 6B), presumably due to the increase in BMP signaling. Indeed, Fam134C$-/-$ organoids exhibited significantly higher levels of BMPR1A (Fig. 6B) as well as increased mRNA levels of Id1 and Id2 (Fig. 6C), which were inversely correlated with the decreased levels of Lgr5 and Olfm4 (being associated with intestinal stemness) in Fam134C$-/-$ organoids (Fig. 6D). To explore the influence of the BMP antagonist Noggin, a crucial component of the organoid media (Sato et al, 2009), we assessed the response of the organoids to various concentrations of Noggin. Notably, increasing Noggin could counteract the attenuating effect of Fam134C ablation on crypt budding of the Fam134C$-/-$ organoids (Fig. 6E,F). Moreover, Noggin treatment restored the elevation of the stem cell marker Lgr5 and reversed the induction of Id1 in the Fam134C$-/-$ organoids, with 500 ng/ml of Noggin being sufficient to achieve an effect comparable to wild-type organoids without treatment (Fig. 6G,H). Collectively, these findings suggest that FAM134C modulates intestinal stem cell maintenance and function by inhibiting BMP signaling.

## Discussion

In this study, we report that FAM134C negatively regulates BMP signaling. We demonstrate that FAM134C inhibits BMP signaling by interacting with BMPR1A and promoting its degradation through the autophagy-lysosome pathway (Fig. 6I). This represents a unique mechanism for the regulation of BMP signaling, in parallel to the previously reported regulation of BMP receptor signaling through the ubiquitin-proteasome system.

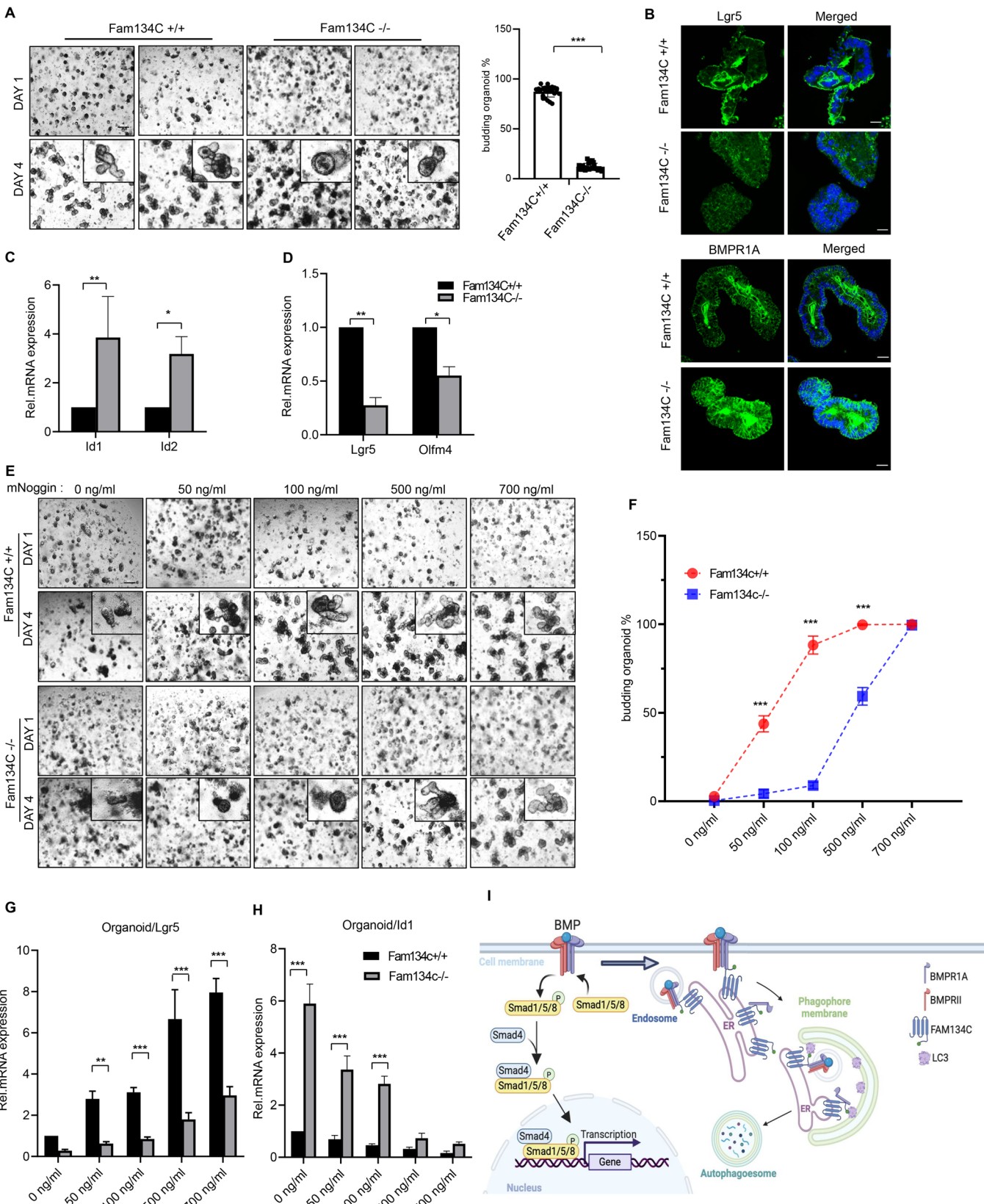

**Figure 6.   Block BMP signaling help budding organoid in FAM134c−/− mice.**

(A) FAM134C deficiency attenuates organoid formation. Organoids were cultured in the medium containing 100 ng/ml of Noggin. Morphology of intestine organoids from WT and FAM134C knockout mice after 1 and 4 days are shown. Right, quantification of organoid budding percentage at 4 days (scale bar, 200 μm). Statistical analysis by unpaired two-tailed Student's *t*-test; *$p < 0.05$, **$p < 0.01$, ***$p < 0.001$; mean ± SD ($n = 5$ mice per genotype; each with five technical replicates). WT vs KO $p = 2.93\text{E-}26$. (B) FAM134C deficiency causes the disappearance of Lgr5 and increased BMPR1A. Immunofluorescence was performed using organoid sections at day 4 from WT and FAM134C knockout mice (scale bar, 50 μm). DAPI (blue fluorescence) was used to stain nuclei. Images are representative of $n = 3$ mice per genotype. (C) Quantitative RT-PCR analysis of Id1 and Id2 in intestinal organoids at day 4. Statistical analysis by unpaired two-tailed Student's *t*-test; *$p < 0.05$, **$p < 0.01$, ***$p < 0.001$; mean ± SD ($n = 3$ independent experiments). Id1 expression WT vs KO $p = 4.22\text{E-}02$; Id2 expression WT vs KO $p = 5.89\text{E-}03$. (D) Quantitative RT-PCR analysis of Lgr5 and Olfm4 in intestinal organoids at day 4. Statistical analysis by unpaired two-tailed Student's *t*-test; *$p < 0.05$, **$p < 0.01$, ***$p < 0.001$; mean ± SD ($n = 3$ independent experiments). Lgr5 expression WT vs KO $p = 6.46\text{E-}05$; Id2 expression WT vs KO $p = 6.95\text{E-}04$. (E, F) Noggin counteracts FAM134C deficiency during organoid formation. Panel E, morphology of intestine organoids from WT and FAM134C knockout mice after treatment with noggin at the indicated concentration after 1 and 4 days (scale bar, 200 μm). (F) Quantification of organoid budding percentage at 4 days in the intestine organoids. Statistical analysis by unpaired two-tailed Student's *t*-test; *$p < 0.05$, **$p < 0.01$, ***$p < 0.001$; mean ± SD ($n = 3$ mice per genotype; each with ten technical replicates). In 50 ng/ml WT vs KO $p = 1.54\text{E-}45$; In 100 ng/ml WT vs KO $p = 4.10\text{E-}61$; In 500 ng/ml WT vs KO $p = 2.11\text{E-}46$. (G, H) Quantitative RT-PCR analysis of Lgr5 (G) and Id1 (H) in the intestine organoids of WT and FAM134C knockout mice after treatment with Noggin at the indicated concentration on day 4. Statistical analysis by unpaired two-tailed Student's *t*-test; *$p < 0.05$, **$p < 0.01$, ***$p < 0.001$; mean ± SD ($n = 3$ independent experiments). For (G), In 50 ng/ml WT vs KO $p = 6.03\text{E-}04$; In 100 ng/ml WT vs KO $p = 9.97\text{E-}05$; In 500 ng/ml WT vs KO $p = 4.46\text{E-}03$; In 1000 ng/ml WT vs KO $p = 4.19\text{E-}04$. For (H), In 0 ng/ml WT vs KO $p = 3.45\text{E-}04$; In 50 g/ml WT vs KO $p = 1.04\text{E-}03$; In 100 ng/ml WT vs KO $p = 1.64\text{E-}04$. (I) A working model for the effect of FAM134C in BMP signaling. In cells, FAM134C interacts with BMPR1A and promotes its degradation through the autophagy-lysosome pathway. Source data are available online for this figure.

## BMPR1A is degraded through the autophagy-lysosome pathway

The BMP pathway is a crucial signaling pathway involved in various biological processes such as development, tissue home-ostasis, and disease (McCarthy et al, 2020; Wang and Chen, 2018; Zhang and Que, 2020). The regulation of BMP signaling is complex and involves both intracellular and extracellular mechanisms. Previous studies have extensively described the role of the ubiquitin-proteasome system in degrading BMP receptors. In particular, the Smurf family E3 ligases antagonize the TGF-β/BMP pathway by targeting TGF-β/BMP receptors or Smads for ubiquitination-mediated degradation (Kavsak et al, 2000; Lin et al, 2000; Zhang et al, 2001; Zhu et al, 1999). However, the involvement of the autophagy-lysosome pathway in regulating BMP receptors has been less clear. A previous study reported that chloroquine has been shown to enhance BMPR2 levels on the cell surface (Dunmore et al, 2013), whereas increased autophagic flux is associated with reduced BMPR2 levels in patients with pulmonary arterial hypertension (PAH) (Gomez-Puerto et al, 2019). Yet, the mechanism underlying BMPR2 stability is unknown. Our work establishes that FAM134C, an ER-resident protein, mediates the degradation of the type I receptor BMPR1A through the autophagy-lysosome pathway, thereby introducing a previously unrecognized mode of BMP receptor regulation. FAM134C directly interacts with BMPR1A and pulls BMPR1A, likely together with BMPRII, into the LC3-containing autophagosomes through its LIR region. FAM134C-mediated reduction in the BMPR1A level can be prevented chemically by autophagy inhibitors or genetically by deleting key autophagy factors like Atg5 or Beclin1. Both in vitro and in vivo evidence support the direct interaction of FAM134C with BMPR1A, leading to its degradation via the autophagy-lysosome pathway.

Consistent with amino acid withdrawal attenuating BMP signaling in cultured cells, fasting can also lead to the disappearance of BMPR1A in the mouse intestinal tissues. The knockout of FAM134C increases the levels of BMPR1A in the cultured cells and the intestinal tissues. Notably, Fam134C-ablated mice are resistant to fasting-induced degradation of BMPR1A. This is accompanied by enhanced phosphorylation of Smad1/5/8, increased Id1 and Id2

expression, and decreased expression of Sox9 at the crypt base, indicating reduced stemness of intestinal stem cells in Fam134C−/− mice. Along the crypt-villus axis in the normal colon, a low level of BMP and a high level of Wnt maintain the stem cell niche at the base of the crypts, while increased BMP and decreased Wnt drive epithelial cell differentiation towards the luminal surface (Sato et al, 2009). Thus, increased BMP signaling in the crypt base suppresses the stemness of intestinal stem cells, in agreement with previous studies (He et al, 2004; Qi et al, 2017). Moreover, reduced stemness in the Fam134C−/− crypt is further confirmed using organoid cultures. The Fam134C-deleted organoids showed increased BMP signaling, leading to reduced numbers of Lgr5-positive stem cells, which can be reversed by increasing the concentration of Noggin in the culture media. These findings provide evidence that BMP signaling is indeed upregulated in Fam134C−/− tissues due to increased expression of BMPR1A.

## FAM134C is a specific regulator of BMP signaling

Members of the FAM134 family reside in the ER and contribute to ER remodeling and quality control. Among them, FAM134B is the first identified mammalian ER-phagy receptor (Khaminets et al, 2015), and its physiological and pathological roles have been investigated in relatively more detail (Mo et al, 2020). Dysfunction or mis-expression of FAM134B impairs ER homeostasis and causes human diseases (Chen et al, 2022). Like FAM134B, FAM134A and FAM134C also reside in the ER and are essential for maintaining ER morphology (Reggio et al, 2021). They share the same amino acid sequence in their LIRs (Khaminets et al, 2015), and may have shared and unique functions. For example, all three members are involved in Collagen I quality control, yet FAM134A appears to function constitutively in the process. However, by proteomic analysis using FAM134 knockout cells, FAM134B and FAM134C have functional overlaps in degrading ER proteins.

FAM134C is a ubiquitously expressed and more abundant protein within the FAM134 family (Wang et al, 2019), indicating a potentially broader role for FAM134C. Our discovery that FAM134C can degrade BMPR1A, and other type I receptors, and likely together with BMPRII, greatly expands the functions of FAM134C beyond ER-phagy. The specificity of FAM134C towards

BMPR1A degradation arises from its strong interaction with BMPR1A, together with the failure of FAM134A and FAM134B to degrade BMPR1A, highlights the unique role of FAM134C in regulating BMP signaling. In line with this specificity, FAM134C does not target the newly synthesized pool of BMPR1A in the ER for degradation through its ER-phagy activity, but instead it recruits membrane-bound BMPR1A into the autophagolysosome. Further investigations are needed to reveal more unique functions of FAM134C in other biological processes, and determine whether specific determinants or post-translational modifications play critical roles in their non-ER-phagy functions. A previous study reports that the activity of FAM134C is regulated by protein kinase CK2 in response to starvation. During starvation, inhibition of mTORC1 blocks CK2-induced FAM134C phosphorylation, hence promoting receptor activation and ER-phagy (Di Lorenzo et al, 2022). Consistent with this observation, our results demonstrate that mTORC inhibition restrains BMP signaling.

Overall, the findings presented in this study underscore the crucial role of FAM134C in mediating a non-ER-phagy function and provide mechanistic insights into its regulation of BMPR1A degradation through lysosomes. These results suggest that targeting FAM134C may hold promise as a potential therapeutic strategy for diseases associated with dysregulated BMP signaling.

# Methods

## Reagents and tools table

| Reagent/resource | Reference or source | Identifier or Catalog Number |
| --- | --- | --- |
| **Experimental models** | | |
| U2OS | Maintained in the Feng lab since 2013 (originally from ATCC) | RRID: CVCL_0042 |
| HEK293T | Maintained in the Feng lab since 2013 (originally from ATCC) | RRID: CVCL_0063 |
| Hela | Maintained in the Feng lab since 2013 (originally from ATCC) | RRID: CVCL_0030 |
| A549 | Maintained in the Feng lab since 2013 (originally from ATCC) | RRID: CVCL_0023 |
| C57BL/6 genetic FAM134C−/− mice | Gift from Xiamen University | N/A |
| **Recombinant DNA** | | |
| pBobi-puro-FAM134C | This study | N/A |
| pBobi- GFP-puro-FAM134C | This study | N/A |
| pLKO.1-shRNA-puro-FAM134C | This study | N/A |
| pcDNA3-YN-FAM134C | This study | N/A |
| pcDNA3-YN-LC3 | This study | N/A |
| pcDNA3-YN-BMPR1A | This study | N/A |
| pcDNA3-YC-FAM134C | This study | N/A |
| pcDNA3-YC-BMPR1A | This study | N/A |

| Reagent/resource | Reference or source | Identifier or Catalog Number |
| --- | --- | --- |
| pRK5-FAM134C | This study | N/A |
| Id1-luc | Gift from Peter ten Dijke | N/A |
| pRK-Smad5-HA | This study | N/A |
| pRK-BMPR1a-HA | This study | N/A |
| pRK-KDEL-streptavidin | This study | N/A |
| pRK-BMPR1A-SBP-GFP | This study | N/A |
| **Antibodies** | | |
| anti-BMPR1A | Abcam | Cat# ab38560 |
| anti-FAM134C | Abcam | Cat# ab202125 |
| anti-MUC2 | Abcam | Cat# ab272692 |
| anti-Chromogranin A | Abcam | Cat# ab254557 |
| anti-SOX9 | Abcam | Cat# ab185966 |
| anti-LGR5 | Abclone | Cat# A12327 |
| anti-LC3B | Abclone | Cat# A19665 |
| anti-HA | Cell Signaling Technology | Cat# 3724 |
| anti-Lamp1 | Cell Signaling Technology | Cat# 9091 |
| anti-Smad1 | Cell Signaling Technology | Cat# 6944 |
| anti-phospho-Smad1(Ser463/465)/Smad5(Ser463/465)/Smad9(Ser465/467) | Cell Signaling Technology | Cat# 13820 |
| anti-mTOR | Cell Signaling Technology | Cat# 2983 |
| anti-phospho-mTOR(Ser2448) | Cell Signaling Technology | Cat# 2971 |
| anti-p62/SQSTM | Proteintech | Cat# 18420-1-AP |
| anti-phospho-Smad1/Smad5/Smad8(S463/465) | Millipore | Cat# AB3848-I |
| anti-Id1 | Santa Cruz | Cat# sc-133104 |
| anti-Myc | Santa Cruz | Cat# sc-40 |
| anti-FLAG M2 | Sigma | Cat# F3165 |
| anti-glyceraldehyde-3-phosphate dehydrogenase (anti-GAPDH) | Sigma | Cat# G8795 |
| Fluor 488-labeled goat anti-rabbit Alexa | Invitrogen | Cat# A-11008 |
| Fluor 488-labeled goat anti-mouse Alexa | Invitrogen | Cat# A-11001 |
| Alexa Fluor 546-labeled goat anti-rabbit | Invitrogen | Cat# A-11035 |
| Alexa Fluor 546-labeled goat anti-mouse | Invitrogen | Cat# A-11030 |
| Goat anti-rabbit IgG (H + L) Secondary antibody, HRP conjugate | Invitrogen | Cat# 31460 |
| Goat anti-mouse IgG (H + L) Secondary antibody, HRP conjugate | Invitrogen | Cat# 31430 |
| **Oligonucleotides and other sequence-based reagents** | | |
| h-FAM134C forward | This study | TTCTGCTGTCCTACTTGATGCT |
| h-FAM134C reverse | This study | ATCCTGTTCCAAATCACGATTGT |

| Reagent/resource | Reference or source | Identifier or Catalog Number |
|---|---|---|
| h-BMPR1A forward | This study | CCTGTTGTCATAGGTCCGTTTT |
| h-BMPR1A reverse | This study | ATCCTGTTCCAAATCACGATTGT |
| h-Id1 forward | This study | CTGCTCTACGACATGAACGG |
| h-Id1 reverse | This study | GAAGGTCCCTGATGTAGTCGAT |
| h- Id2 forward | This study | GCTATACAACATGAACGACTGCT |
| h- Id2 reverse | This study | AATAGTGGGATGCGAGTCCAG |
| h-GAPDH forward | This study | AGCCACATCGCTCAGACAC |
| h-GAPDH reverse | This study | GCCCAATACGACCAAATCC |
| h-ATG5 forward | This study | AAAGATGTGCTTCGAGATGTGT |
| h-ATG5 reverse | This study | AGGTGTTTCCAACATTGGCTC |
| h-BECN1 forward | This study | GGTGTCTCTCGCAGATTCATC |
| h-BECN1 reverse | This study | TCAGTCTTCGGCTGAGGTTC |
| m-Id1 forward | This study | AGAACCGCAAAGTGAGCAAGGT |
| m-Id1 reverse | This study | GGTGGTCCCGACTTCAGACT |
| m-Id2 forward | This study | ATCCCACTATCGTCAGCCTGCAT |
| m-Id2 reverse | This study | ATTCAGATGCCTGCAAGGACAGG |
| m-ALP forward | This study | AGAAGTTCGCTATCTGCCTTGCCT |
| m-ALP reverse | This study | TGGCCAAAGGGCAATAACTAGG |
| m-BMPR1A forward | This study | ATGCTCCATGGCACTGGTATGA |
| m-BMPR1A reverse | This study | GGCAGTGTCCTGAGCAATAGCA |
| m-BMPR1B forward | This study | AGATTGGAAAAGGCCGCTATG |
| m-BMPR1B reverse | This study | GATGTCAACCTCATTTGTGTC |
| m-BMPRII forward | This study | GAGGACTGGCTTATCTTCAC |
| m-BMPRII reverse | This study | AGCTCCTTCTAGCACTTCTG |
| m-Lgr5 forward | This study | CGGGACCTTGAAGATTTCCT |
| m-Lgr5 reverse | This study | GATTCGGATCAGCCAGCTAC |
| m-Oflm4 forward | This study | CGAGACTATCGGATTCGCTATG |
| m-Oflm4 reverse | This study | TTGTAGGCAGCCAGAGGGAG |
| m-Smad1 forward | This study | GTTCAGGCAGTTGCTTACGA |
| m-Smad1 reverse | This study | ATTGTTGGACGGATCTGTGA |
| m-Smad6 forward | This study | CTCCGGGTGAATTCTCAGAT |
| m-Smad6 reverse | This study | TGGTCGTACACCGCATAGAG |
| m-Smad7 forward | This study | GGGCTTTCAGATTCCCAACTT |
| m-Smad7 reverse | This study | AGGGCTCTTGGACACAGTAGA |
| m-BMP2 forward | This study | ACCCGCTGTCTTCTAGCGT |
| m-BMP2 reverse | This study | TTTCAGGCCGAACATGCTGAG |
| m-BMP4 forward | This study | TTGATACCTGAGACCGGGAAG |
| m-BMP4 reverse | This study | ACATCTGTAGAAGTGTCGCCTC |
| m-Gemilin1 forward | This study | CTGGGGACCCTACTGCCAA |
| m-Gemilin1 reverse | This study | TTTGCACCAATCTCGCTTCAG |
| SYBR Green Master Mix | Applied Biosystems | Cat# 4368708 |
| **Chemicals, enzymes, and other reagents** | | |
| BMP2 | R&D | Cat# 355-BM-100 |
| BMP4 | Stem RD | Cat# BMP4-100 |
| EGF | Invitrogen | Cat# PHG0311 |
| Recombinant mouse Noggin | R&D | Cat# 1967-NG-025 |
| Recombinant mouse R-spondin1 | R&D | Cat# 7150-RS-025 |
| Dorsomorphin | Sigma-Aldrich | Cat# P5499 |

| Reagent/resource | Reference or source | Identifier or Catalog Number |
|---|---|---|
| puromycin | Sigma-Aldrich | Cat# 540411 |
| AHA | Sigma-Aldrich | Cat# 900892 |
| MG132 | Selleckchem | Cat# S2619 |
| CQ | Selleckchem | Cat# S4430 |
| rapamycin | Selleckchem | Cat# S1039 |
| MHY1485 | MedChemExpress | Cat# HY-B0795 |
| Torin1 | MedChemExpress | Cat# HY-13003 |
| 3-MA | MedChemExpress | Cat# HY-19312 |
| Benzohase endonuclease | Sigma-Aldrich | Cat# E1014 |
| EcoRI | New England Biolabs | Cat# R3101L |
| SalI | New England Biolabs | Cat# R3138L |
| BamHI | New England Biolabs | Cat# R3136S |
| XbaI | New England Biolabs | Cat# R0145L |
| fetal bovine serum | ExCell Bio | Cat# FSP500 |
| DMEM | Corning | Cat# 10-013-CV |
| RPMI1640 | Corning | Cat# 10-040-CV |
| ENR | R&D | Cat# HSC005 |
| X-tremeGENE HP DNA Transf.Reag. | Roche Applied Science | Cat# 6366236001 |
| PEI | Polyscience | Cat# 23966-1 |
| TnT® Quick Coupled Transcription/ Translation System | Promega | Cat# L2080 |
| Dual-Luciferase®Reporter Assay System | Promega | Cat# E1960 |
| protein A Sepharose | GE Healthcare | Cat# 17-0963-03 |
| protease inhibitor cocktail | Sigma-Aldrich | Cat# P2714-1BTL |
| alkene-Biotin | Sigma-Aldrich | Cat# 764213 |
| Quick start Bradford Dye Reagent | Bio-Rad | Cat# 500-0205 |
| ECL detection reagent | Thermo Fisher | Cat# 32106 |
| Streptavidin agarose beads | Thermo Fisher | Cat# 20347 |
| methionine-free DMEM | Gibico | Cat# 21013024 |
| TRIzol reagent | Invitrogen | Cat# 15596018CN |
| DAPI reagent | Invitrogen | Cat# P36971 |
| LysoTracker™ Blue DND-22 | Invitrogen | Cat# L7525 |
| Tissue-Tek® O.C.T. Compound | Sakura | Cat# 4583 |
| **Software** | | |
| ABI PRISM 7500 Sequence Detector System | Applied Biosystems | |
| ImageJ/Fiji | NIH | https://imagej.nih.gov/ |
| Prism 9.0 | Graphpad | https://www.graphpad.com/ |
| **Other** | | |
| Pierce™ Cell Surface Protein Biotinylation and Isolation Kit | Thermo Scientific | Cat# A44390 |

| Reagent/resource | Reference or source | Identifier or Catalog Number |
|---|---|---|
| ER Enrichment Kit | Invent Biotechnologies | Cat# ER-036 |
| Lysosome Isolation Kit | Invent Biotechnologies | Cat# LY-034 |
| Endosome Isolation and Cell Fractionation Kit | Invent Biotechnologies | Cat# ED-028 |
| Click-iT™ Protein Reaction Buffer Kit | Thermo Fisher | Cat# C10276 |
| PrimeScript® RT reagent kit | TaKaRa | Cat# RR036A |
| Zeiss LSM880 confocal microscope | Carl Zeiss | |
| vibrating blade microtome | Microm | HM650 |
| Vectra 3 Automated Quantitative Pathology Imaging System | Akoya Biosciences | |

## Methods and Protocols

### Plasmids

Expression plasmids for HA-tagged Smad5 and HA-tagged BMPR1A have been previously described (Duan et al, 2006; Lin et al, 2006). Full-length FAM134C, its mutant, and deletion mutants were obtained by PCR and cloned into the EcoRI and SalI sites of pRK5F (derivative of pRK5, Genentech). FAM134C was also subcloned into the BamHI and XbaI sites of lentiviral vectors pBobi-puro or pBobi-GFP-puro. DNA containing the FAM134C shRNA cassette was cloned into the pLKO.1-shRNA-puro vector according to standard methods. The sequence integrity of all plasmids was confirmed by sequencing. Reporter gene plasmid Id1-luc was kindly provided by Peter ten Dijke. The RUSH system consists of two expression plasmids: an ER hook (KDEL) fused to streptavidin (SA), and a streptavidin-binding peptide (SBP) linked to green fluorescent protein (GFP) and BMPR1A (BMPR1A-SBP-GFP).

### Cell cultures and transfection

U2OS, HEK293T, and HeLa cells were cultured in DMEM (Corning, Mediatech, Manassas, VA) supplemented with 10% fetal bovine serum (FBS) (ExCell Bio) at 37 °C in a humidified incubator with 5% $CO_2$. A549 cells were maintained in RPMI1640 (Corning) with 10% FBS. U2OS and A549 cells were transfected with X-tremeGENE (Roche Applied Science), and HEK293T cells with PEI (Polyscience), respectively.

### Lentivirus production and stable cell line generation

A lentiviral plasmid for expression of either FAM134C or its shRNA was transfected into HEK293T cells together with packaging plasmid psPAX2 and envelope plasmid pMD2.G. After 48 h, supernatants containing lentiviral particles were collected and infected into host cells. Stable cells were selected in the presence of 1 ng/ml of puromycin (Sigma).

### Immunoprecipitation and Western blotting

HEK293T cells were transfected with pRK5F-FAM134C and other plasmids as indicated in the figures, and harvested 24 h after transfection. Co-IP was carried out using an anti-FLAG antibody and protein A Sepharose (GE Healthcare). After three times of washes, immunoprecipitated proteins were eluted in SDS loading buffer, separated by SDS-PAGE, transferred onto PVDF membranes (Millipore), and detected by Western blotting analysis. For Western blotting, cells were incubated with or without reagents or transfected with plasmid as indicated in the figures, washed twice with ice-cold PBS and suspended in ice-cold lysis buffer (50 mM Tris•HCl pH 7.4, 1% Triton X-100, 1% sodium deoxycholate, 0.1% SDS, 150 mM NaCl, 1 mM EDTA, and 1 mM sodium orthovanadate) containing a protease inhibitor cocktail (Sigma-Aldrich). The protein concentration was determined using the Bradford assay (Bio-Rad). Protein was solubilized in sample buffer at 95 °C for 5 min and resolved by 10% SDS-PAGE. For immunoblotting, proteins were electro-transferred onto a PVDF membrane and blocked with 5% nonfat milk in Tris-buffered saline (TBS) solution containing 0.10% Tween 20 (TBST) at room temperature for 1 h. Then, the membrane was incubated with the primary antibody at 4 °C overnight. After washing with TBST, the blot was incubated with secondary antibody (1:10000, Sigma-Aldrich) for 1 h at room temperature. After washing, the antibody-bound antigen was detected with the ECL detection reagent (Thermo Fisher).

### In vitro protein binding assay

GST fusions of BMPR1A (ICD) proteins were prepared from the *Escherichia coli* strain DE3. In vitro translation of FAM134C or FAM134B was carried out by using the Quick Coupled transcription/translation system (Promega). In vitro binding was carried out by using GST-tagged BMPR1A (ICD) incubated with in vitro-translated FAM134C or FAM134B deletion mutants for 2 h in binding buffer (0.5% NP-40, 150 mM NaCl, 50 mM Tris-HCl, and 5 mM EDTA) and monitored by Western blotting.

### Isolation of membrane, ER, endosome, and lysosome proteins

Membrane: Pierce™ Cell Surface Protein Biotinylation and Isolation Kit (Thermo Scientific, A44390) was used to isolate membrane protein according to the manufacturer's instructions. Briefly, $1 \times 10^7$ cells are first labeled with Sulfo-NHS-SS-Biotin, a thiol-cleavable amine-reactive biotinylation reagent. Cells are subsequently lysed with detergent, and the labeled proteins are then isolated with Avidin Agarose. The agarose-bound labeled proteins are released by reduction of the disulfide bond with 10 mM DTT.

ER, endosome, and lysosome: ER Enrichment Kit (Invent Biotechnologies, ER-036), Lysosome Isolation Kit (Invent Biotechnologies, LY-034), and Endosome Isolation and Cell Fractionation Kit (Invent Biotechnologies, ED-028) were used to isolate respective organelles according to the manufacturer's instructions.

### Retention using selective hooks (RUSH) transport assay

Retention using selective hooks (RUSH) transport assay were established as previously described (25, 26). U2OS cells were cultured in a 35 mm glass-bottom dish (for live imaging) or six-well plate and transfected with BMPR1A-RUSH with other indicated plasmids for 24 h. The release of the RUSH reporter was stimulated by the addition of biotin at a final concentration of 50 μM for 1 h. Images were collected under a Zeiss LSM880 confocal microscope (Carl Zeiss). Protein lysates were analyzed by Western blotting.

## BONCAT assay

In the BONCAT assay, cells were first cultured with methionine-free DMEM (Cat# 21013024, Gibco) for 1 h and then labeled with 50 μM non-canonical amino acid azidohomoalanine (AHA, Cat# 900892, Sigma-Aldrich) for 12 h. The newly synthesized protein carries an active azide (N3) group. Total cells were then lysed with lysis buffer (1% SDS, 50 mM Tris, pH 8.0) including protease inhibitors and Benzonase endonuclease (Cat# E1014, Sigma-Aldrich). The collected supernatants were incubated with 100 μM alkene-Biotin (Cat# 764213, Sigma-Aldrich) with Click-iT™ Protein Reaction Buffer Kit (Cat# C10276, Thermo Fisher). With this click reaction, pulse-labeled newly synthesized proteins were covalently attached to the biotin tag, purified using Streptavidin agarose beads (Cat# 20347, Thermo Fisher). After washing, the immunoprecipitated proteins were subjected to immunoblotting to examine the newly synthesized proteins.

## Luciferase reporter assays

A reporter plasmid for Id1-luciferase (Id1-luc), previously described (Korchynskyi and ten Dijke, 2002), was used to measure BMP-induced transcription. Cells were transfected with reporter plasmids, together with a Renilla luciferase plasmid to normalize the transfection efficiency. Briefly, 24 h after transfection, cells were treated with BMP2 (20 ng/ml) for 12 h, and luciferase activity was measured by using a Dual-Luciferase reporter assay system (Promega). All assays were carried out in triplicate and normalized against Renilla luciferase activity.

## RT-PCR analysis

To determine transcript levels, total RNAs were obtained by TRIzol reagent (Invitrogen) (Xiao et al, 2024). RNAs were reverse transcribed to complementary DNA using the PrimeScript® RT reagent kit (TaKaRa). Quantitative reverse transcriptase (qRT)-PCR was performed on an ABI PRISM 7500 Sequence Detector System (Applied Biosystems) using gene-specific primers and SYBR Green Master Mix (Applied Biosystems).

## Bimolecular fluorescence complementation (BiFC) assay

The BiFC assay was carried out as previously described (Dai et al, 2009). YN (aa 1–155 fragment of Venus, a variant of YFP)-LC3, YN-FAM134C, YN-BMPR1A, and YC (aa 155–238 fragment of YFP)-FAM134C, YC-BMPR1A fusions were constructed in the pcDNA3 expression vector and expressed in U2OS or A549 cells. Fluorescence signals were examined under a Zeiss LSM880 confocal microscope (Carl Zeiss).

## Immunofluorescence and immunohistochemistry

For cell immunostaining, cells were cultured on glass coverslips with or without stimulation for different time periods as indicated in the figure legends. After washing twice with PBS, cells were incubated with 4% PFA for 15 min and then incubated with 1x PBS/0.3% Triton containing 5% normal goat serum for 1 h at room temperature. Then, the cells were exposed to primary antibody at 4 °C overnight, rinsed three times with PBS, and incubated with Fluor 488- or Alexa Fluor 546-labeled goat anti-rabbit or anti-mouse Alexa (1:1000, Invitrogen) for 1.5 h at room temperature. After three washing steps, all the slides and coverslips were mounted with ProLong Gold antifade with DAPI reagent (Invitrogen). Images were collected under a Zeiss LSM880 confocal microscope.

For Lysotracker Blue DND-22 staining, probes (1 μM, Invitrogen) were directly added to the medium, incubated for 30 min, washed with fresh medium, and then imaged by confocal microscopy.

For immunofluorescence on frozen sections, intestinal tissues isolated from mice were washed with cold PBS, fixed with 4% formaldehyde solution, and embedded in Tissue-Tek® OCT Compound (Sakura). The sections were prepared with a vibrating blade microtome (HM650, Microm) and permeabilized and blocked with PBST solution (5% BSA and 2% Triton X-100 in PBS) for 1 h at room temperature. For organoid staining, isolated organoids were first fixed for 1 h with 4% paraformaldehyde at room temperature, permeabilized for 20 min with 0.2% Triton X-100 on ice, and blocked in 5% BSA for 1 h at room temperature. The sections or organoids were then incubated overnight with the primary antibody at 4 °C. The following primary antibodies were used: goat anti-Chromogranin A (ChgA), rabbit anti-p-Smad1/5/8, rabbit anti-BMPR1A, and rabbit anti-Lgr5. The fluorescein-labeled secondary antibodies (Life Technologies, 1:1000) were applied for 1 h at room temperature. Confocal laser scanning was done under a Zeiss LSM880 confocal microscope.

For immunohistochemistry, formalin-fixed, paraffin-embedded intestine sections (5 μm) were de-paraffinized in isopropanol and graded alcohols, followed by antigen retrieval, and endogenous peroxidase quenched by $H_2O_2$. Sections were then blocked with 5% FBS for 30 min and incubated overnight with rabbit anti-Sox9 and rabbit anti-BMPR1A at 4 °C. Secondary horseradish peroxidase-conjugated anti-rabbit antibody (Invitrogen, 1:200) was added for 2 h, followed by development with DAB+ chromogen according to the manufacturer's (Dako Cytomation) recommendations. Slides were counterstained with Mayer's haematoxylin, dehydrated, and mounted with Eukitt Quick-hardening. The slides were captured using the Vectra 3 Automated Quantitative Pathology Imaging System (Akoya Biosciences).

## Mice

Fam134c−/− mice were kindly provided by Xiamen University. Mice were back-crossed into the C57BL/6 genetic background for at least ten generations. Both male and female mice, ranging from 2 to 4 months old in age were used. No statistical method was used to predetermine sample size. Generally, we used at least five mice per genotype in each experiment. None of the animals was excluded from the experiment, and the animals used were not randomized. The investigators were not blinded to allocation during experiments and outcome assessment. All animal studies were performed in accordance with the relevant guidelines and under the approval of the Institutional Animal Care and Use Committee of our institute.

For irradiation, mice received a single dose of abdominal X-ray radiation (10 Gy) and were then analyzed at different time points. The order of animal handling and testing was also randomized to avoid systematic bias during outcome assessment. All animal studies were performed in accordance with the relevant guidelines and under the approval of the Institutional Animal Care and Use Committee of our institute.

## Isolation of intestinal crypts and organoid culture

Intestinal crypts were isolated and cultured as previously described (Meng et al, 2022; Zhang et al, 2020). Briefly, the mouse intestine was cut longitudinally and washed three times with cold PBS. Villi were carefully scraped away, and small pieces (5 mm) of intestine were incubated in 2 mM EDTA in PBS for 40 min on ice. These pieces were then vigorously suspended in cold PBS, and the mixture was passed through a 70-μm cell strainer (BD Biosciences). The

crypt fraction was enriched through centrifugation (3 min at $300–400 \times g$). Then the crypts were embedded in Matrigel (BD Biosciences) and seeded on a 48- or 24-well plate. After polymerization, crypts were placed in the ENR medium (DMEM/F12 supplemented with Penicillin/Streptomycin, GlutaMAX-I, N2, B27, and N-acetylcysteine, which was further supplemented with 50 ng/ml EGF, 100 ng/ml Noggin, and 500 ng/ml R-spondin1) and refreshed every 2 days. For BMP4 treatment, organoids were cultured in the ENR medium for 2.5 days, followed by culturing in ENR medium or ERB (EGF, R-spondin1, and 10 ng/ml BMP4; R&D) medium for the indicated time. For Noggin withdrawal, organoids were cultured in ENR medium for 2.5 days, followed by passaging. ENR medium or ER medium was added for the indicated time, respectively. For passaging, the organoids embedded in Matrigel in each well were directly suspended in 1 ml cold PBS after removal of medium and were pelleted by centrifugation (3 min at $300–400 \times g$). The pelleted organoids were embedded in fresh Matrigel and seeded on a plate followed by the addition of culture medium as indicated in the figure legends.

### Experimental design and blinding

Due to the requirement for daily cage-side treatment adjustments, investigators could not be blinded to group allocation. To minimize bias, data collection and histological scoring were carried out by a researcher not involved in the original group assignment.

### Statistical analysis

Data were provided as means ± SEM; $n$ represents the number of independent experiments. Data were tested for significance using the unpaired Student's $t$-test or ANOVA as appropriate. Differences were considered statistically significant when $p$ values were <0.05. Statistical analysis was performed with GraphPad Prism 9, San Diego, California, USA; www.graphpad.com.

### Graphics

The Synopsis image was created with BioRender.com.

## Data availability

This study includes no data deposited in external repositories.

The source data of this paper are collected in the following database record: biostudies:S-SCDT-10_1038-S44318-025-00581-3.

## Peer review information

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

## Acknowledgements

We thank Peter ten Dijke for Id1-luc, Xiaochen Wang for advice on the RUSH assay, and Qirou Wu for her technical support in mouse organoid experiments. We are indebted to the Core Facility of Life Sciences Institute and its staff for assisting with a variety of cell and biochemical analyses. We thank members of the Feng Laboratory for helpful discussion and technical assistance. This research was partly supported by grants from the National Key Research and Development Program of China (2022YFC3401500), the National Natural Science Foundation of China (U21A20356, 32321002, 31730057, and 31771546), the Natural Science Foundation of Zhejiang Province (LD21C070001 and LZ22C070001), Hangzhou West Lake Pearl Project Leading Innovative Youth Team Project (TD2023017), and the Fundamental Research Funds for the Central Universities.

## Author contributions

**Shuchen Gu**: Data curation; Formal analysis; Funding acquisition; Writing—original draft. **Hanchenxi Zhang**: Resources; Data curation. **Jin Cao**: Methodology. **Zhou He**: Data curation. **Jianfeng Wu**: Resources. **Xia Liu**: Methodology. **Mingjie Zheng**: Data curation. **Ting Liu**: Resources; Supervision. **Bin Zhao**: Supervision. **Pinglong Xu**: Supervision. **Qiming Sun**: Resources; Validation. **Jianping Jin**: Supervision. **Xia Lin**: Supervision. **Yi Yu**: Supervision. **Jiahuai Han**: Resources. **Xin-Hua Feng**: Conceptualization; Supervision; Funding acquisition; Writing—review and editing.

Source data underlying figure panels in this paper may have individual authorship assigned. Where available, figure panel/source data authorship is listed in the following database record: biostudies:S-SCDT-10_1038-S44318-025-00581-3.

## Disclosure and competing interests statement

The authors declare no competing interests.

# Expanded View Figures

**Figure EV1.  FAM134C degrades BMPR1A by autophagy.**

(**A, B**) Quantitative RT-PCR analysis of FAM134C and BMPR1A mRNAs in U2OS cells stably expressing FAM134C-FLAG or carrying an empty vector control. mean ± SD ($n$ = 3 independent experiments). (**C**) Expression of FAM134C and GAPDH were measured in wildtype (WT) or FAM134C-KO U2OS cells by Western blotting. (**D**) Quantitative RT-PCR analysis of BMPR1A mRNA in WT or FAM134C-KO U2OS cells. mean ± SD ($n$ = 3 independent experiments). (**E**) HEK293T cells were co-transfected with vectors encoding FAM134C-FLAG and ALK1-HA, ALK2-HA, ALK3-HA, ALK6-HA, or BMPR2-HA, then FAM134C was detected by anti-FLAG antibody and the protein ALK1, ALK2, ALK3, ALK6, or BMPR2 was detected by HA antibody. (**F**) Quantitative RT-PCR analysis of ATG5 mRNA in U2OS with stable knockdown of ATG5 or control. mean ± SD ($n$ = 3 independent experiments). (**G**) Quantitative RT-PCR analysis of BECN1 mRNA in U2OS with stable knockdown of BECN1 or control. mean ± SD ($n$ = 3 independent experiments). (**H**) BECN1 downregulation blocks the degradation of BMPR1A by FAM134C. U2OS cells with BECN1 stable knockdown were transfected with FAM134C-FLAG or the FLAG vector. Twenty-four hours after transfection, cells were treated with BMP2 (50 ng/ml) for 0.5 or 1 h. Levels of BMPR1A, p-Smad1/5/8, Smad1, FAM134C, and GAPDH were measured by Western blotting. Source data are available online for this figure.

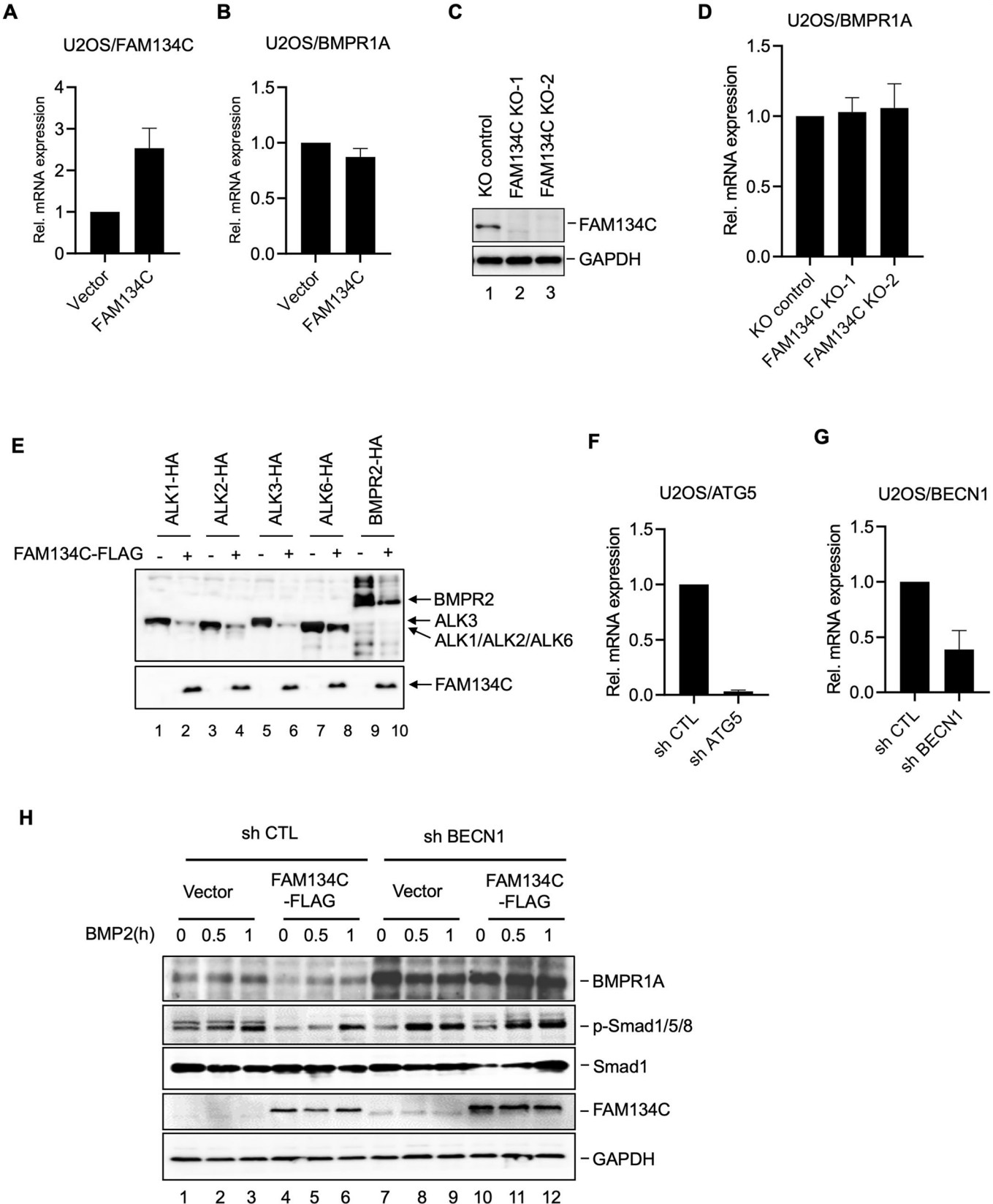

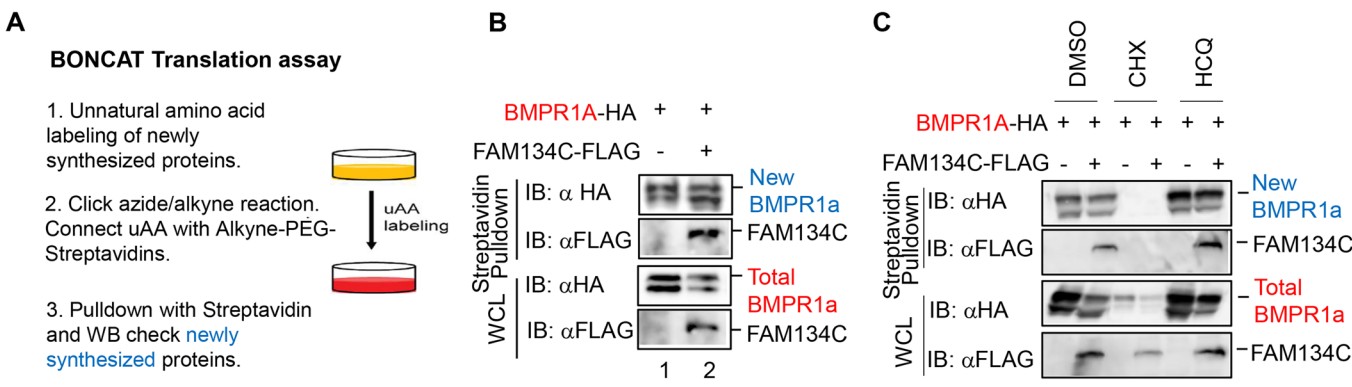

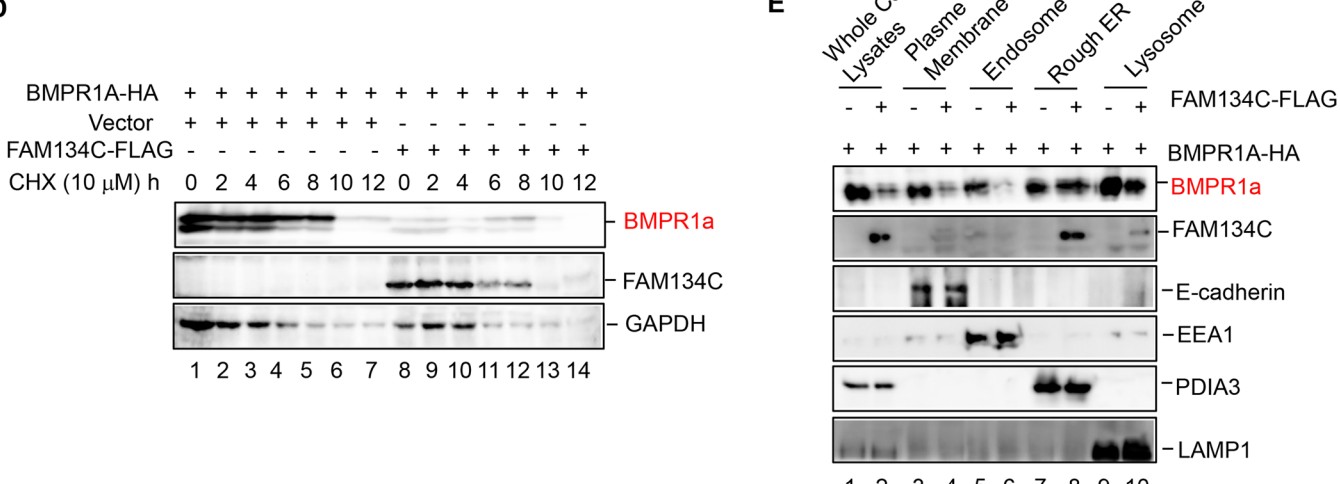

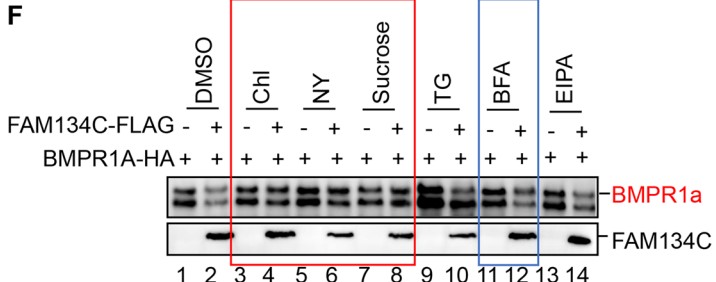

| Inhibitors | Mode of actions |
|---|---|
| **Chlorpromazine (Chl)** | Blocks Clathrin-mediated endocytosis pathway. |
| **Nystatin (NY)** | Blocks Caveolae-mediated endocytosis pathway. |
| **Sucrose** | Inhibits receptor-mediated endocytosis due to its effects on osmotic conditions and membrane fluidity. |
| **Brefeldin A (BFA)** | Blocks the transport of secreted proteins and membrane proteins from the endoplasmic reticulum to the Golgi apparatus. |
| **Ethyl-isopropyl amiloride (EIPA)** | Blocks macropinocytosis. |
| **Thapsigargin (TG)** | An endoplasmic reticulum (ER) stress inducer. |

**Figure EV2.  FAM134C targets membrane-bound BMPR1A for degradation.**

(**A**) Schema of the BONCAT translation assay. (**B**, **C**) FAM134C does not degrade newly synthesized BMPR1A. HEK293T cells were co-transfected with FAM134C-FLAG and BMPR1A-HA and/or empty vector control for 24 h. In (**C**), transfected cells were further treated with cycloheximide (10 μM) or HCQ (50 nM) for 8 h. The newly synthesized protein was collected by Streptavidin agarose beads. BMPR1A expression was detected by Western blotting. (**D**) FAM134C overexpression causes BMPR1A degradation. HEK293T cells were co-transfected with FAM134C-FLAG and BMPR1A-HA, and treated with cycloheximide (CHX, 10 mM) for the indicated times. Expression of BMPR1A, FAM134C, and GAPDH were measured by Western blotting. (**E**) FAM134C reduces the levels of BMPR1A on the plasma membrane and endosomes, but not the ER. HCQ blocks BMPR1A degradation in the lysosome. Organelles were collected and analyzed for BMPR1A by Western blotting in HEK293T cells expressing FAM134C-FLAG and BMPR1A-HA. Isolation of the plasma membrane, endosome, ER, and lysosome is described in the Materials and Methods section. Expression of BMPR1A, FAM134C, E-cadherin (membrane marker), EEA1 (endosome marker), PDIA3 (ER marker), and LAMP1 (lysosome marker) were measured by Western blotting. (**F**) FAM134C degrades internalized BMPR1A. HEK293T cells were co-transfected with FAM134C-FLAG and BMPR1A-HA, and treated with Chlorpromazine (Chl, 50 nM), Nystatin (NY, 50 nM), Thapsigargin (TG, 500 mM), or ethyl-isopropyl amiloride (EIPA, 10 mM) for 15 h, sucrose (0.2 M) and Brefeldin A (BFA, 10 mg/ml) for 2 h. Levels of BMPR1A and FAM134C-FLAG were measured by Western Blotting. (**G**) Inhibitors used in (**F**) above. Source data are available online for this figure.

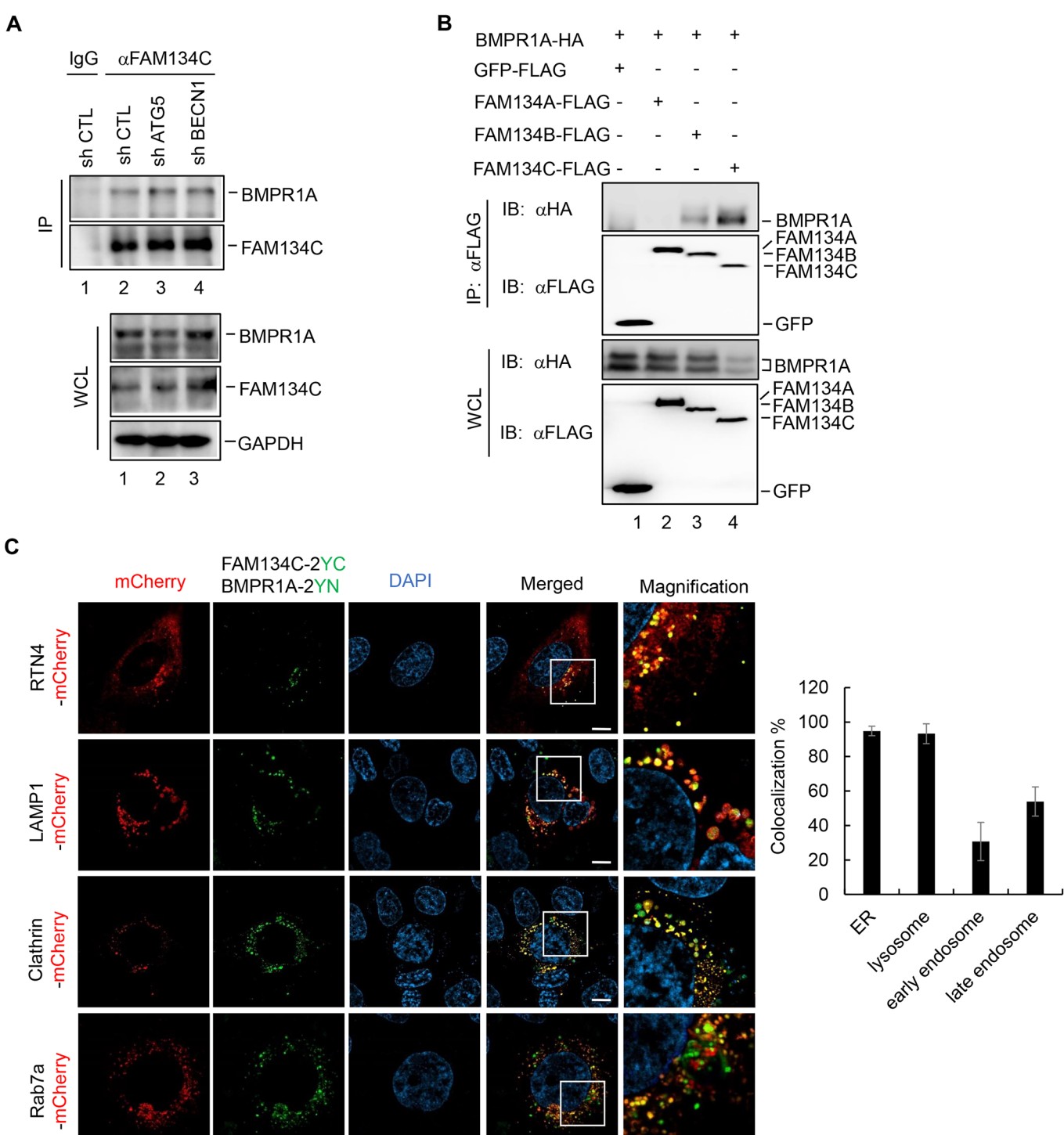

**Figure EV3. FAM134C interacts with BMPR1A.**

(A) ATG5 or BECN1 downregulation had no effect on the interaction between FAM134C and BMPR1A. ATG5 or BECN1 were stably knocked down in U2OS cells, and then FAM134C was immunoprecipitated by anti-FAM134C and anti-BMPR1A antibodies was detected by Western blotting. (B) BMPR1A interacts with FAM134C and to a lesser extent FAM134B, but not FAM134A. HEK293T cells were transfected with vectors encoding BMPR1A-HA and FAM134A-FLAG, FAM134B-FLAG, or FAM134C-FLAG, and then FAM134A/B/C were immunoprecipitated by anti-FLAG antibody and FAM134-bound BMPR1A was detected by HA antibody. (C) FAM134C is colocalized with BMPR1A. U2OS cells were co-transfected with RTN4-mCherry, LAMP1-mCherry, Clathrin-mCherry, or Rab7a-mCherry together with YC-tagged FAM134C and YN-tagged BMPR1A. Cells were treated with HCQ for 2 h and then analyzed using Zeiss LSM880. The green YFP fluorescence signal indicates the FAM134C-BMPR1A interaction. DAPI (blue fluorescence) was used to stain nuclei (scale bar, 5 μm). The percentage of the FAM134C-BMPR1A interaction colocalized with an organelle was quantified by ImageJ. Bar and error bars show the mean ± SD ($n = 3$ independent experiments; each with three technical replicates). Source data are available online for this figure.

Times:     0:01:62          0:01:81          0:01:81          0:02:22

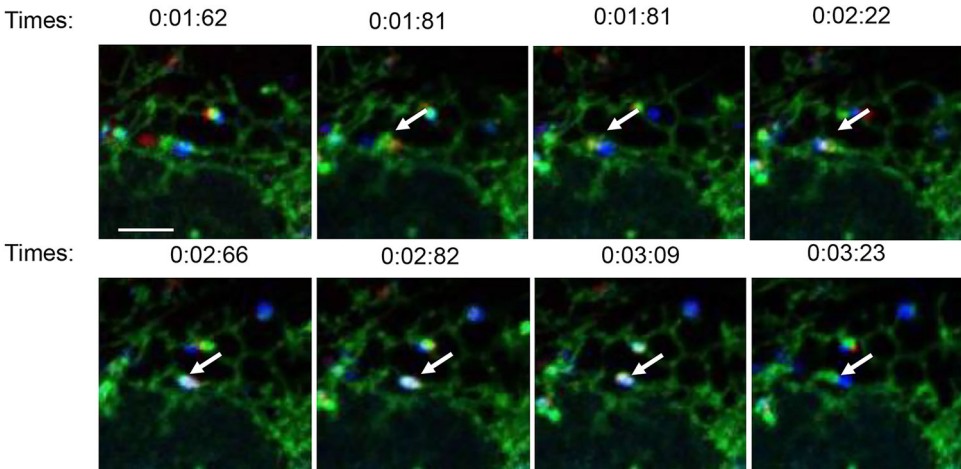

Times:     0:02:66          0:02:82          0:03:09          0:03:23

**Figure EV4.  FAM134C targets BMPR1A into autophagosomes through LC3.**

Live cell image shows that FAM134C targets membrane-bound BMPR1A to lysosomes for degradation. FAM134C-KO U2OS cells stably express FAM134C-GFP and BMPR1A-mCherry. Lysotracker (blue fluorescence) was used to stain the lysosome. Cells were analyzed under Zeiss LSM880 (scale bar, 1 μm). Source data are available online for this figure.

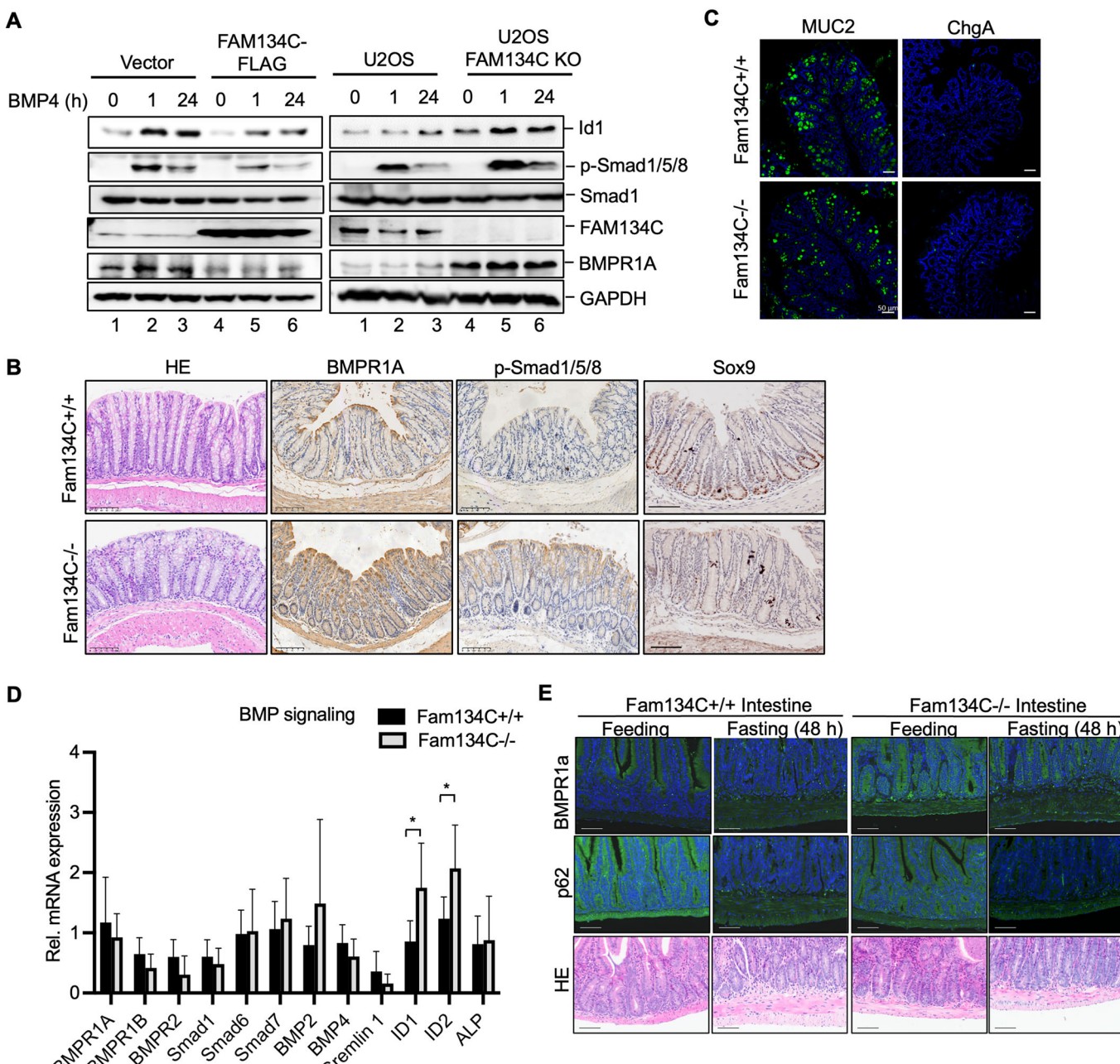

**Figure EV5.  Lack of FAM134C increases BMP signaling in mouse colons.**

(A) FAM134C inhibits BMP responses. U2OS cells stably expressing FAM134C-FLAG or with FAM134C knockout or parental cells were treated with BMP (50 ng/ml) for 1 and 24 h. Levels of Id1, p-Smad1/5/8, Smad1, FAM134C, BMPR1A, and GAPDH were measured by Western blotting. (B) Hematoxylin-eosin staining and immunohistochemical staining of BMPR1A, p-Smad1/5/8 and Sox9 in colons of WT and FAM134C knockout mice (scale bar, 50 mm). Colon sections were collected at week 8. Images are representative of *n* = 6 mice per genotype. (C) Immunohistochemical staining of MUC2 and ChgA in colons of WT and FAM134C knockout mice. Colon sections were collected at week 8 (scale bar, 100 mm). Images are representative of *n* = 6 mice per genotype. (D) Quantitative RT-PCR analysis of the indicated genes in the colons of WT and FAM134C knockout mice. Statistical analysis by unpaired two-tailed Student's *t*-test; *$p < 0.05$, **$p < 0.01$, ***$p < 0.001$; mean ± SD (*n* = 3 mice per genotype; each with two technical replicates). Id1 expression, WT vs KO $p = 2.36\text{E-}02$; Id2 expression, WT vs KO $p = 2.87\text{E-}02$. (E) Immunofluorescence staining of BMPR1A and p62, and Hematoxylin-eosin staining in small intestines of WT and FAM134C knockout mice with or without fasting. Proximal jejunum sections were made at week 8. DAPI (blue fluorescence) was used to stain nuclei (scale bar, 100 mm). Source data are available online for this figure.

