## [Peer Review File · The EMBO Journal]

Reticulophagy receptor FAM134C restrains BMP receptor signaling

Shuchen Gu, Hanchenxi Zhang, Jin Cao, Zhou He, Jianfeng Wu, Xia Liu, Mingjie Zheng, Ting Liu, Bin Zhao, Pinglong Xu, Qiming Sun, Jianping Jin, Xia Lin, Yi Yu, Jiahuai Han, and Xin-Hua Feng

Corresponding author(s): Xin-Hua Feng (xhfeng@zju.edu.cn)

Review Timeline:

Submission Date:	20th Dec 24
Editorial Decision:	3rd Feb 25
Revision Received:	13th Jul 25
Editorial Decision:	22nd Aug 25
Revision Received:	27th Aug 25
Accepted:	9th Sep 25

Editor: William Teale

Transaction Report:

Dear Prof. Feng,

Thank you again for the submission of your manuscript entitled "Reticulophagy receptor FAM134C restrains BMP receptor signaling" and for your patience during the review process. We have now received the reports from the referees, which I copy below.

As you can see from their comments, both referees ask for you to go into greater detail: Referee #1 is particularly concerned that you clarify the relationship between FAM134C-mediated BMP breakdown and ER-phagy, whereas Referee #2 is keen to see more physiological context for this regulation. That said, both of them point out that your work is timely and interesting.

At this stage, I am considering this manuscript as a better fit for EMBO Reports; however, it would certainly be a good idea to discuss this on a Zoom call; please indicate when a convenient time would be, once you have had time to digest both reports. In the meantime, I am happy to formally invite you to address the comments of all referees in a revised version of the manuscript. That said, I must emphasise that, depending on the revisions that are feasible within a timeframe of three to six months, I may recommend publication in EMBO Reports after re-review.

I should add that it is The EMBO Journal policy to allow only a single major round of revision and that it is therefore important to resolve the main concerns at this stage. I believe the concerns of the referees are reasonable and addressable, but please contact me if you have any questions, need further input on the referee comments or if you anticipate any problems in addressing any of their points. Please, follow the instructions below when preparing your manuscript for resubmission.

I would also like to point out that as a matter of policy, competing manuscripts published during this period will not be taken into consideration in our assessment of the novelty presented by your study ("scooping" protection). We have extended this 'scooping protection policy' beyond the usual 3 month revision timeline to cover the period required for a full revision to address the essential experimental issues. Please contact me if you see a paper with related content published elsewhere to discuss the appropriate course of action.

Again, please contact me at any time during revision if you need any help or have further questions.

Thank you very much again for the opportunity to consider your work for publication. I look forward to your revision.

Best regards,

William Teale

William Teale, Ph.D.
Editor
The EMBO Journal

When submitting your revised manuscript, please carefully review the instructions below and include the following items:

- 1) a .docx formatted version of the manuscript text (including legends for main figures, EV figures and tables). Please make sure that the changes are highlighted to be clearly visible.
- 2) individual production quality figure files as .eps, .tif, .jpg (one file per figure).
- 3) a .docx formatted letter INCLUDING the reviewers' reports and your detailed point-by-point response to their comments. As part of the EMBO Press transparent editorial process, the point-by-point response is part of the Review Process File (RPF), which will be published alongside your paper.
- 4) a complete author checklist, which you can download from our author guidelines ([https://wol-prod-cdn.literatumonline.com/pb-assets/embo-site/Author Checklist%20-%20EMBO%20J-1561436015657.xlsx](https://wol-prod-cdn.literatumonline.com/pb-assets/embo-site/Author%20Checklist%20-%20EMBO%20J-1561436015657.xlsx)). Please insert information in the checklist that is also reflected in the manuscript. The completed author checklist will also be part of the RPF.
- 5) Please note that all corresponding authors are required to supply an ORCID ID for their name upon submission of a revised manuscript.

6) We require a 'Data Availability' section after the Materials and Methods. Before submitting your revision, primary datasets produced in this study need to be deposited in an appropriate public database, and the accession numbers and database listed under 'Data Availability'. Please remember to provide a reviewer password if the datasets are not yet public (see <https://www.embopress.org/page/journal/14602075/authorguide#datadeposition>). If no data deposition in external databases is needed for this paper, please then state in this section: This study includes no data deposited in external repositories. Note that the Data Availability Section is restricted to new primary data that are part of this study.

Note - All links should resolve to a page where the data can be accessed.

8) For data quantification: please specify the name of the statistical test used to generate error bars and P values, the number (n) of independent experiments (specify technical or biological replicates) underlying each data point and the test used to calculate p-values in each figure legend. The figure legends should contain a basic description of n, P and the test applied. Graphs must include a description of the bars and the error bars (s.d., s.e.m.).

9) We would also encourage you to include the source data for figure panels that show essential data. Numerical data can be provided as individual .xls or .csv files (including a tab describing the data). For 'blots' or microscopy, uncropped images should be submitted (using a zip archive or a single pdf per main figure if multiple images need to be supplied for one panel). Additional information on source data and instruction on how to label the files are available at .

10) We replaced Supplementary Information with Expanded View (EV) Figures and Tables that are collapsible/expandable online (see examples in <https://www.embopress.org/doi/10.15252/embj.201695874>). A maximum of 5 EV Figures can be typeset. EV Figures should be cited as 'Figure EV1, Figure EV2' etc. in the text and their respective legends should be included in the main text after the legends of regular figures.

12) Our journal encourages inclusion of *data citations in the reference list* to directly cite datasets that were re-used and obtained from public databases. Data citations in the article text are distinct from normal bibliographical citations and should directly link to the database records from which the data can be accessed. In the main text, data citations are formatted as follows: "Data ref: Smith et al, 2001" or "Data ref: NCBI Sequence Read Archive PRJNA342805, 2017". In the Reference list, data citations must be labeled with "[DATASET]". A data reference must provide the database name, accession number/identifiers and a resolvable link to the landing page from which the data can be accessed at the end of the reference. Further instructions are available at .

13) In order to increase the reproducibility and reach of your work, The EMBO Journal includes a table of reagents that were used in the study. Please provide this along with your revisions.

Further instructions for preparing your revised manuscript:

We realize that it is difficult to revise to a specific deadline. In the interest of protecting the conceptual advance provided by the work, we recommend a revision within 3 months (4th May 2025). Please discuss the revision progress ahead of this time with the editor if you require more time to complete the revisions. Use the link below to submit your revision:

Referee #1:

In the manuscript entitled "Reticulophagy receptor FAM134C restrains BMP receptor signaling," the authors demonstrate that FAM134C interacts with BMPR1A, enabling its sequestration into LC3-containing autophagosomes for subsequent degradation. Blocking autophagy or FAM134C binding to LC3 inhibits BMPR1A degradation and enhances BMP signaling. The *in vivo* data support this claim, suggesting a role for autophagy in restraining BMP signaling. The manuscript provides intriguing insights into the role of FAM134C in BMP receptor signaling. However, significant gaps remain in the proposed mechanism and in the quality of the data.

Main Comments: The data clearly demonstrate that BMPR1A is degraded via FAM134C-mediated autophagy. However, the authors propose that this process is ER-phagy independent, which has not been demonstrated. In my opinion, this degradation likely occurs via ER-phagy, and the fraction of BMPR1A targeted for degradation is the portion produced in the ER and not yet secreted. The authors need to clearly demonstrate where the receptor is sequestered by FAM134C.

Quantifications are notably missing from most of the experiments. The authors must appropriately report results, including quantification, bar graph data, number of experiments, number of cells counted, and statistical analyses. This applies to both immunofluorescence and western blotting.

Additional Comments:

1. mTORC1 is also activated by growth factors, so it is unclear why growth factor starvation increased Id1-luc activation and BMP2 levels. The rationale for this observation needs to be clarified. mTORC1 activity cannot be accurately measured by assessing p-mTOR levels. Instead, the phosphorylation of canonical substrates such as p4EBP1 and pS6K should be analyzed.
2. Rapamycin decreases BMPR1A levels, suggesting that additional receptors may play a role. The authors should compare the

effects of silencing different FAM134 proteins, either individually or in combination, on BMPR1A levels. Despite the authors' claims, Supplementary Figure 2 clearly shows that FAM134B interacts with BMPR1A.

3. The BiFC assay is not informative for protein localization. The observed large dots have uncertain localization and specificity. Quantification is missing, and co-staining with organelle markers has not been performed. The quality and interpretability of these data are questionable.

4. The immunofluorescence data in Figure 4E are difficult to interpret. There are no clear LC3-GFP dots, and the GFP signal has a diffuse/reticular pattern. Once again, there is no quantification of the data.

5. Physiological Relevance of Autophagy in BMP Signaling: The in vitro data suggest that autophagy must be activated by starvation to degrade BMPR1A. This concept links autophagy to BMP signaling in an intriguing way. To assess the physiological relevance of this observation, the authors should analyze BMPR1A levels in control and knockout mice upon fasting in selected tissues. The authors have not emphasized this aspect.

6. The sentence, "BMPs play a pivotal role in maintaining the balance between homeostatic self-renewal and proliferation of intestinal epithelial cells," requires references and a more detailed explanation. Specifically, the inhibitory role of BMP signaling on intestinal stem cells needs to be clearly articulated.

Referee #2:

In the present study, the authors identified FAM134C, an ER-localized protein, as a selective autophagy receptor for BMPR1A. FAM134C bridges BMPR1A and LC3, leading to degradation of BMPR1A and subsequent downregulation of BMP signaling. They further show in vivo evidence for the FAM134C-mediated repression of BMP signaling, focusing on the intestine. This is a very nice work reporting a novel mode of regulation of BMP signaling, though the physiological significance of this regulatory system currently remains unclear. Overall, their conclusions were well supported by technically-sound biochemical/cell biological analyses. I have several minor comments as follows.

Minor points:

1. BMP signaling is involved in homeostasis of a wide variety of organs/tissues. However, in this study, the authors focused only on the intestine of FAM134C knockout mice. How about other organs/tissues that are affected by excess BMP signaling? Is the effect of FAM134C knockout specific to the intestine?
2. FAM134C overexpression attenuates BMP signaling (Fig. 4A). Therefore, FAM134C appears to induce degradation of functional BMPR1A, but not misfolded BMPR1A, in the ER and inhibit BMP signaling. Does FAM134C randomly induce degradation of newly synthesized BMPR1A in the ER? Alternatively, does the degradation require some modification of BMPR1A (for example, dimer formation with BMPR2 etc. See comment 3.)? Discussion on this subject in the contexts of biological processes would be helpful for readers (though the authors briefly mentioned it: line 353-).
3. Is there any advantage in using selective autophagy to downregulate BMPR1A, instead of ubiquitin-dependent proteasomal degradation? Include some discussion.
4. A question on the specificity of this regulatory system: Does FAM134C also downregulate other BMP type I receptors, BMPR1B, ACVR1, and ACVRL1?
5. Fig. 2B: Though FAM134C did not directly interact with BMPR2, co-transfection of FAM134C and BMPR2 resulted in downregulation of BMPR2. This should be mentioned and discussed.
6. Fig. 2C: BMPR1A (ECD) is smaller than BMPR1A (ICD), but labels on the right-side show oppositely.
7. Figure 5A, legend: State that the medium contains 100 ng/ml of Noggin, for readers' convenience.
8. Insert scale bars into cell images (Fig. 1I, 2F, 3B, 3E, 5B). No information on scale bars in the legend to Fig. 3A.
9. Line 314: BMPR 2 > BMPR2
10. Line 703: MHY1486 > 1485. //

Manuscript EMBOJ-2024-119994

We thank the reviewers for their positive reviews and valuable feedback on our manuscript. We carefully followed the reviewers' instructions, conducting the recommended experiments and undertaking additional efforts to perform supplementary experiments. Through this revised manuscript, we have integrated new data, rectified errors, and clarified descriptions where necessary. We are confident that the revised manuscript effectively addresses all raised queries and has been strengthened significantly. The corresponding modifications have been implemented in the revised paper, with major changes highlighted such as in the text with burgundy font color and in the figure panel numbering (e.g. **A** in yellow).

Point-by-point reply to the reviewers' comments:**Reviewer #1:**

Main Comments: *The data clearly demonstrate that BMPR1A is degraded via FAM134C-mediated autophagy. However, the authors propose that this process is ER-phagy independent, which has not been demonstrated. In my opinion, this degradation likely occurs via ER-phagy, and the fraction of BMPR1A targeted for degradation is the portion produced in the ER and not yet secreted. The authors need to clearly demonstrate where the receptor is sequestered by FAM134C.*

Response: Using multiple methods, we confirmed that hat FAM134C does not inhibit secretion of newly synthesized BMPR1A via degradation. The results are shown as below:

1). To rule out that ER is the compartment where BMPR1A is degraded (via ER-phagy), we monitored its trafficking at the secretory pathway using RUSH system (PMID: 22406856; 33796610). In the RUSH assay, BMPR1A is tagged with GFP and bound to a streptavidin binding peptide (SBP), and a streptavidin molecule bound to KDEL (ER retention signal) (Panel A, below). In the absence of biotin, BMPR1A is retained in the ER, but in the presence of biotin, the BMPR1A-GFP-SBP is released from ER to the ultimate compartment (e.g. plasma membrane) (Panel B, below).

In Panel C (below), we observed that FAM134C overexpression reduced the level of BMPR1A-RUSH (due to degradation) in the presence of biotin, but it had little effect in the absence of biotin. This suggests that FAM134C induces degradation of biotin-released BMPR1A-RUSH at plasma membrane, not the BMPR1A forcefully retained in ER. The data is now added as **Figure 2A-C** to the revised manuscript.

Fig. 2A

2B

2C

2). Using bio-orthogonal non-canonical amino acid tagging (BONCAT) to label newly synthesized BMPR1A, we observed that FAM134C overexpression reduced the total level of BMPR1A (due to degradation), but did not affect newly synthesized BMPR1A levels. These data are now shown in new **Figure 2D**, and new supplemental **Figure S2A-S2C**.

Fig. S2A

BONCAT Translation assay

1. Unnatural amino acid labeling of newly synthesized proteins.

2. Click azide/alkyne reaction. Connect uAA with Alkyne-PEG-Streptavidins.

3. Pulldown with Streptavidin and WB check newly synthesized proteins.

Fig. S2B

293T Transient expressed BMPR1A

Fig. 2D

A549 Endogenous BMPR1A

Fig. S2C

293T Transient expressed BMPR1A

3). Following cycloheximide (CHX) treatment to inhibit new protein synthesis, BMPR1A degradation was accelerated in cells overexpressing FAM134C, but decelerated in FAM134C knockout cells. These findings indicate that FAM134C influences the degradation of “old” pool of, rather than newly-synthesized, of BMPR1A. These data are now shown in new **Figure 2E, 2F**, and new supplemental **Figure S2D**.

Fig. 2E

U2OS

Fig. 2F

U2OS

Fig. S2D

4). Subcellular fractionation revealed that FAM134C degrades BMPR1A in the plasma membrane and endosomes, but not significantly in the endoplasmic reticulum (ER). Hydroxychloroquine (HCQ) treatment resulted in significant BMPR1A accumulation in lysosomes, confirming that FAM134C mediates BMPR1A degradation via lysosomal autophagy. These data are now shown in new **Figure 2G**, and new supplemental **Figure S2E**.

Fig. 2G

Fig. S2E

5). Using various inhibitors, we found that chlorpromazine (chl), nystatin (Ny), and sucrose (hypertonic media), but not brefeldin A (BFA), block FAM134C-mediated degradation of BMPR1A. This result suggests that while FAM134C does not prevent transport of newly synthesized BMPR1A, it negatively impacts the level of internalized BMPR1A. These data are now shown in new **Figure 2H**, and new supplemental **Figure S2F, G**.

Fig. S2G

Inhibitors	Mode of actions
Chlorpromazine (Chl)	Blocks Clathrin-mediated endocytosis pathway.
Nystatin (NY)	Blocks Caveolae-mediated endocytosis pathway.
Sucrose	Inhibits receptor-mediated endocytosis due to its effects on osmotic conditions and membrane fluidity.
Brefeldin A (BFA)	Blocks the transport of secreted proteins and membrane proteins from the endoplasmic reticulum to the Golgi apparatus.
Ethyl-isopropyl amiloride (EIPA)	Blocks macropinocytosis.
Thapsigargin (TG)	An endoplasmic reticulum (ER) stress inducer.

6). Using live cell imaging, we discovered that FAM134C pulls the membrane-bound BMPR1A (likely from the plasma membrane) and subsequently sends it to lysosomes for degradation. This is now shown in new **Figure 2I**, and new video clip **Figure S2H**.

Fig. 2I

Time-lapse images

U2OS FAM134C KO :

pBobi-FAM134C-GFP

BMPR1A-mCherry

Lysotracker blue

Additional Comments:

1. *mTORC1* is also activated by growth factors, so it is unclear why growth factor starvation increased *Id1-luc* activation and *BMP2* levels. The rationale for this observation needs to be clarified. *mTORC1* activity cannot be accurately measured by assessing *p-mTOR* levels. Instead, the phosphorylation of canonical substrates such as *p4EBP1* and *pS6K* should be analyzed.

Response: Serum starvation and growth factor deprivation are not the same conditions. FBS contains over 1,000 different components, including not only growth factors, but also inhibitors of several signaling pathways. We speculate that the presence of BMP inhibitors in serum and/or serum factors that may induce the expression of secreted BMP inhibitors (autocrine) could directly antagonize BMP signaling. In light of this speculation, we consistently observed that serum-starved cells are hypersensitive to BMP. Consequently, we also saw higher levels of BMP-induced phosphorylation of Smad1/5/8 and subsequent transcriptional responses. Because this study does not address how/why FBS withdrawal affects BMP signaling, we decide not to include this part in the revised manuscript.

With regard to the mTOR readout, we now added the blots of p-S6K. As shown in the following blots, AA depletion attenuates BMP-mediated phosphorylation of Smad1/5/8. As expected, AA depletion, which typically inhibits mTOR, can lead to reduction in the level of p-S6K (revised **Figure 1C**).

Fig. 1C

2. Rapamycin decreases *BMPR1A* levels, suggesting that additional receptors may play a role. The authors should compare the effects of silencing different *FAM134* proteins, either individually or in combination, on *BMPR1A* levels. Despite the authors' claims, *Supplementary Figure 2* clearly shows that *FAM134B* interacts with *BMPR1A*.

Response: To address the reviewer's question, we carried out three experiments.

- 1) Overexpression experiments in 293T cells demonstrated that among three FAM134 family members, only FAM134C overexpression led to reduction in the BMPR1A level (Panel A, below, lane 4).
- 2) In A549 cells, we generated sgRNA knockdown cell pools targeting FAM134A, FAM134B, or FAM134C. Notably, only FAM134C knockdown resulted in increased BMPR1A expression (Panel B, below, lane 6 & 7).
- 3) The level of BMPR1A protein was also increased in FAM134C-knockdown U2OS cells, and further rescue experiments showed that only FAM134C re-expression restored BMPR1A degradation (Panel C, below, compare lane 5 to lane 2).

These findings collectively indicate that among the FAM134 family members, only FAM134C possesses the capacity to mediate BMPR1A degradation, although FAM134B interacts with BMPR1A. Our current work specifically establishes FAM134C as one such mediator of autophagic BMPR1A degradation. As FAM134B regulates ER-phagy, this also implicates that BMPR1A degradation is not through ER-phagy.

3. The BiFC assay is not informative for protein localization. The observed large dots have uncertain localization and specificity. Quantification is missing, and co-staining with organelle markers has not been performed. The quality and interpretability of these data are questionable.

Response: Thanks to the reviewer for his/her comment. I like to point out that BiFC is not intended for localization of a given protein. Yet, the reconstituted fluorescence can indicate the location where the protein-protein interaction occurs (green color in this case). We have used BiFC assays in two figures, i.e. **Figure 3E** and **Figure 4A & B** (previously Figure 2E and Figure 3A & B, respectively).

- 1) **Figure 3E** shows that co-expression of BMPR1A-2YN and FAM134C-2YC enables reconstitution of YFP/GFP green fluorescence in punctate foci, indicating the BMPR1A-FAM134C interaction; this interaction presumably occurs at the ER-membrane contact.

- 2) In **Figure 4A**, the data shows that LC3-2YN and FAM134C-2YC also reconstitute green fluorescence in punctate foci, indicating the LC3-FAM134C interaction.
- 3) The original **Figure 3B** (now revised to become **Figure 4B**) shows that the “green” BMPR1A-FAM134C complex colocalizes with LC3-mCherry in response to starvation. Now in the revised manuscript, we have redone the experiment and also include quantitation of the colocalization.

Furthermore, now in the revised manuscript, we have included the organelle markers. These data are now shown in new **Figure S3C**. Consistent with our conclusion that FAM134C is an ER-resident protein that recruits membrane-bound BMPR1A for lysosomal degradation, we could observe colocalization of the “green” FAM134C-BMPR1A complex initially attached to ER and moved toward lysosomes, particularly upon starvation.

Fig. S3C

4. The immunofluorescence data in Figure 4E are difficult to interpret. There are no clear LC3-GFP dots, and the GFP signal has a diffuse/reticular pattern. Once again, there is no quantification of the data

Response: This is referred to as previous Figure 3E, now **Figure 4E**. We have to apologize that we somehow wrongly labeled the images in the original submission. We have now made the correction and added statistical analysis. It is evident that the colocalization of FAM134C with LC3 is disrupted by the deletion or mutation of the LIR (LC3-interacting region).

5. *Physiological Relevance of Autophagy in BMP Signaling:* The *in vitro* data suggest that autophagy must be activated by starvation to degrade BMPR1A. This concept links autophagy to BMP signaling in an intriguing way. To assess the physiological relevance of this observation, the authors should analyze BMPR1A levels in control and knockout mice upon fasting in selected tissues. The authors have not emphasized this aspect.

Response: Following the reviewer's suggestion, we conducted the recommended experiment in mice. The results from immunofluorescent staining and Western blotting analysis indicate that after 48 h of fasting, BMPR1A levels in the intestinal tissues of wildtype mice significantly decline. In contrast, the knockout of FAM134C restores levels of BMPR1A, which is also broadly distributed in these tissues. These findings have been included in the revised manuscript (**Figure 5F**).

6. The sentence, "BMPs play a pivotal role in maintaining the balance between homeostatic self-renewal and proliferation of intestinal epithelial cells," requires references and a more detailed explanation. Specifically, the inhibitory role of BMP signaling on intestinal stem cells needs to be clearly articulated.

Response: Per the reviewer's suggestion, we have now cited the following references and revised the text accordingly.

Wang S, Chen YG. BMP signaling in homeostasis, transformation and inflammatory response of intestinal epithelium. *Sci China Life Sci.* 2018 Jul;61(7):800-807.

Zhang Y, Que J. BMP signaling in development, stem cells, and diseases of the gastrointestinal tract. *Annu Rev Physiol.* 2020 Feb 10;82:251-273.

McCarthy N, Kraiczy J, Shivdasani RA. Cellular and molecular architecture of the intestinal stem cell niche. *Nat Cell Biol.* 2020 Sep;22(9):1033-1041.

Referee #2

In the present study, the authors identified FAM134C, an ER-localized protein, as a selective autophagy receptor for BMPR1A. FAM134C bridges BMPR1A and LC3, leading to degradation of BMPR1A and subsequent downregulation of BMP signaling. They further show in vivo evidence for the FAM134C-mediated repression of BMP signaling, focusing on the intestine. This is a very nice work reporting a novel mode of regulation of BMP signaling, though the physiological significance of this regulatory system currently remains unclear. Overall, their conclusions were well supported by technically-sound biochemical/cell biological analyses. I have several minor comments as follows.

Response: We would like to express our gratitude to this reviewer for their very positive feedback on our study. We are addressing several of their minor suggestions in the following.

Minor points:

1. BMP signaling is involved in homeostasis of a wide variety of organs/tissues. However, in this study, the authors focused only on the intestine of FAM134C knockout mice. How about other organs/tissues that are affected by excess BMP signaling? Is the effect of FAM134C knockout specific to the intestine?

Response: We have actually examined the steady-state levels of BMPR1A in other tissues/organs such as hearts and bones. The results show that the levels of BMPR1A are also markedly increased in those examined tissues of the FAM134C KO mice. Since the physiological functions of FAM134C in these tissues are under early investigations, these are not included in the current study.

2. FAM134C overexpression attenuates BMP signaling (Fig. 4A). Therefore, FAM134C appears to induce degradation of functional BMPR1A, but not misfolded BMPR1A, in the ER and inhibit BMP signaling. Does FAM134C randomly induce degradation of newly synthesized BMPR1A in the ER? Alternatively, does the degradation require some modification of BMPR1A (for example, dimer formation with BMPR2 etc. See comment 3.)? Discussion on this subject in the contexts of biological processes would be helpful for readers (though the authors briefly mentioned it: line 353-).

Response: This comment aligns with the major concern raised by **Reviewer #1**. To address this issue, we employed five different methods, and our data support the conclusion that FAM134C specifically degrades the membrane-bound pool of BMPR1A (either at the plasma membrane or in endosomes), rather than the newly synthesized pool. Please refer to our response to Reviewer #1 for further details.

Additionally, we investigated whether FAM134C-mediated degradation of BMPR1A involves ubiquitination, a modification typically linked to proteasomal degradation. As demonstrated below, FAM134C does not alter the ubiquitination pattern of BMPR1A, in contrast to Smurf1.

3. Is there any advantage in using selective autophagy to downregulate BMPR1A, instead of ubiquitin-dependent proteasomal degradation? Include some discussion.

Response:

Thanks to the reviewer for her/his question. Our following answer is speculative:

TGF- β superfamily type I receptors undergo internalization and are distributed across both lipid raft and non-raft membrane domains. Clathrin-mediated endocytosis into EEA1-positive endosomes facilitates TGF- β (most likely BMP as well) signaling, whereas lipid raft-caveolar internalization directs the Smad7-Smurf-associated receptors toward rapid degradation (PMID: 12717440, 19050695). BMP type receptors should behave the same.

We speculate that for rapid degradation, Smad7-Smurf complex bind to BMPR1A and promotes the latter's ubiquitin-dependent proteasomal degradation. This process is highly

specific and requires the negative feedback product Smad7.

The endosome may either recycle its contents back to the plasma membrane or direct them to lysosomes for degradation. FAM134C-mediated degradation of BMPR1A occurs in the lysosome and thus may have been through the endosomal pathway. Selective autophagy can degrade larger protein complexes, organelles, or aggregated proteins that are beyond the capacity of the proteasome. Selective autophagy can efficiently degrade the BMPR signaling complexes, ensuring complete removal of the receptor and its associated signaling components, particularly under stress conditions.

The choice between selective autophagy and ubiquitin-dependent proteasomal degradation for downregulating BMPR1A may depend on the specific cellular context, stress conditions, the desired timing of the response.

4. A question on the specificity of this regulatory system: Does FAM134C also downregulate other BMP type I receptors, BMPR1B, ACVR1, and ACVRL1?

Response: We have also investigated the effects of FAM134C on other type I receptors and found that overexpressed FAM134C promotes the degradation of ACVR1, and ACVRL1, and less so of BMPR1B. At present, we do not know why the extent of BMPR1B degradation is notably less pronounced compared to the other receptors (Panel A, below). In addition, BMPRII is also degraded by FAM134C, likely through its complex with type I receptor.

In addition, in the FAM134C-depleted U2OS cells, the steady-state level of BMPR1A is significantly increased, whereas that of BMPR1B is not (Panel B, below).

5. Fig. 2B: Though FAM134C did not directly interact with BMPR2, co-transfection of FAM134C and BMPR2 resulted in downregulation of BMPR2. This should be mentioned and discussed.

Response: Per the reviewer's suggestion, we have now added this to the discussion. We have also noticed that knockdown of FAM134C not only increases the protein level of BMPR1A, but also indeed stabilizes BMPR2.

6. *Fig. 2C: BMPR1A (ECD) is smaller than BMPR1A (ICD), but labels on the right-side show oppositely.*
7. *Figure 5A, legend: State that the medium contains 100 ng/ml of Noggin, for readers' convenience.*
8. *Insert scale bars into cell images (Fig. 1I, 2F, 3B, 3E, 5B). No information on scale bars in the legend to Fig. 3A.*
9. *Line 314: BMRP 2 > BMPR2*
10. *Line 703: MHY1486 > 1485. //*

Response: We sincerely appreciate the reviewer's observations. In response to points 6 to 10, we have made the necessary corrections and modifications as per the reviewer's instructions.

Dear Prof. Feng,

We have now received re-review reports from two referees, which I have included below. As you will see, you have addressed their concerns satisfactorily; however, I would like you to address the remaining points in the discussion section. Before I can finally accept the manuscript, there are some remaining editorial points which need to be addressed. In this regard would you please:

limit the number of keywords to five,
in the reference section, limit longer author lists to ten (using et al.),
include a data availability section,
include a "Disclosure and competing interests statement",
remove the AC/CrediT section from the text,
remove references to 'data not shown'
complete author checklists,
upload both main and EV figures as individual, high-resolution Figure files; Supplemental Figures should be renamed as Figure EV1-EV5 with the appropriate callouts and legends listed below the main figure legends in the manuscript file,
include a Reagents and Tools section,
save source data in a scheme of one figure per folder and then upload them as .zip files. For example, all the Source data files for figure 1 need to be saved in a single folder and this needs to be zipped and then uploaded as "SD figure 1.zip" file. For EV and/or appendix figures, ZIP together all source data. The completed source data checklist should be uploaded as Related Manuscript File,
define the annotated p values ****/****/**/* and provide the exact p-values for the same in the legends of figures 1A, B, I; 3G, H; 4B, F; 5A, B, H-J; 6A, C, D, F, G, H; S5 D, as appropriate,
indicate the statistical test used for data analysis in the legends of figures 1A, B, I; 3G, H; 4B, F; 5A, B, H-J; 6A, C, D, F, G, H; S5 D,
define the nature of n in the legends of figures 1A, B; 4F, 5A, B; 6C, D; S5 D,
define error bars in the legends of figures 1A, B, I; 3G, H; 4B, F; 6A, C, D, F, G, H; S3 C, S5 D,
include a scale bar in figure S2H, and
correct the section order as follows: Title page - Abstract - Keywords - Introduction - Results - Discussion - Methods - Data Availability - Acknowledgements - Disclosure and Competing Interests Statement - References - Figure Legends - Table(s) - Expanded View Figure Legends.

We include a synopsis of the paper (see <http://emboj.embopress.org/>). Please provide me with a general summary image, a two sentence statement and 3-5 bullet points that capture the key findings of the paper.

I am looking forward to receiving your revised manuscript.

EMBO Press is an editorially independent publishing platform for the development of EMBO scientific publications.

Best wishes,

William

William Teale, PhD
Editor
The EMBO Journal
w.teale@embojournal.org

- a point-by-point response to the referees' comments, with a detailed description of the changes made (as a word file).
 - a word file of the manuscript text.
 - individual production quality figure files (one file per figure)
 - a complete author checklist, which you can download from our author guidelines (<https://www.embopress.org/page/journal/14602075/authorguide>).
 - Expanded View files (replacing Supplementary Information)
- Please see out instructions to authors
<https://www.embopress.org/page/journal/14602075/authorguide#expandedview>
- a Reagents and Tools Table as part of the Methods section, which can be downloaded from our author guidelines (<https://www.embopress.org/page/journal/14602075/authorguide#structuredmethods>)

We realize that it is difficult to revise to a specific deadline. In the interest of protecting the conceptual advance provided by the work, we recommend a revision within 3 months (20th Nov 2025). Please discuss the revision progress ahead of this time with the editor if you require more time to complete the revisions. Use the link below to submit your revision:

Referee #1:

The authors have performed several new experiments that support their model, and I am overall satisfied with their revision. However, it is important that they clearly define and describe their model. For example, Fig. 6I shows that FAM134C forms a complex with BMPR1A, but it is depicted as a soluble protein, which is in contrast with its known nature as a membrane-embedded protein. There are examples of ER proteins (e.g., RTN3) that promote ER-PM contacts and the endocytosis of growth factor receptors (PMID: 28495747). Are the authors proposing a similar model? It will be important to clarify the model through which internalization would occur, keeping in mind the nature of the proteins involved.

Referee #2:

In the present study, the authors identified FAM134C, an ER-localized protein, as a selective autophagy receptor for BMPR1A. FAM134C bridges BMPR1A and LC3, leading to degradation of BMPR1A and subsequent downregulation of BMP signaling. They further show *in vivo* evidence for the FAM134C-mediated repression of BMP signaling, focusing on the intestine. This is a very nice work reporting a novel mode of regulation of BMP signaling, though the physiological significance of this regulatory system currently remains unclear. Overall, their conclusions were well supported by technically-sound biochemical/cell biological analyses.

In this revised manuscript, the authors have addressed all of my concerns. However, I have one comment on a newly added passage.

Line 404-406, "Our discovery that FAM134C can degrade BMPR1A, and other type I receptors and likely together with BMPRII, greatly expands the functions of FAM134C beyond ER-phagy." This may be confusing to readers because they did not present any data on other type I receptors and BMPRII (rebuttal letter, page15). The data should be presented as supplemental figures.

All editorial and formatting issues were resolved by the authors.

Dear Prof. Feng,

I am pleased to inform you that your manuscript has been accepted for publication in the EMBO Journal.

Congratulations to you and your team!

Yours sincerely,

William

William Teale, PhD
Editor
The EMBO Journal
w.teale@embojournal.org
